# Symbolic Discovery of Optimization Algorithms

**Xiangning Chen**[1][2][§][*]    **Chen Liang**[1][§]    **Da Huang**[1]    **Esteban Real**[1]

**Kaiyuan Wang**[1]    **Hieu Pham**[1]    **Xuanyi Dong**[1]    **Thang Luong**[1]

**Cho-Jui Hsieh**[2]    **Yifeng Lu**[1]    **Quoc V. Le**[1]

[§]Equal & Core Contribution

[1]Google                [2]UCLA

## Abstract

We present a method to formulate algorithm discovery as program search, and apply it to discover optimization algorithms for deep neural network training. We leverage efficient search techniques to explore an infinite and sparse program space. To bridge the large generalization gap between proxy and target tasks, we also introduce program selection and simplification strategies. Our method discovers a simple and effective optimization algorithm, **Lion** (*EvoLved Sign Momentum*). It is more memory-efficient than Adam as it only keeps track of the momentum. Different from adaptive optimizers, its update has the same magnitude for each parameter calculated through the sign operation. We compare Lion with widely used optimizers, such as Adam and Adafactor, for training a variety of models on different tasks. On image classification, Lion boosts the accuracy of ViT by up to 2% on ImageNet and saves up to 5x the pre-training compute on JFT. On vision-language contrastive learning, we achieve 88.3% *zero-shot* and 91.1% *fine-tuning* accuracy on ImageNet, surpassing the previous best results by 2% and 0.1%, respectively. On diffusion models, Lion outperforms Adam by achieving a better FID score and reducing the training compute by up to 2.3x. For autoregressive, masked language modeling, and fine-tuning, Lion exhibits a similar or better performance compared to Adam. Our analysis of Lion reveals that its performance gain grows with the training batch size. It also requires a smaller learning rate than Adam due to the larger norm of the update produced by the sign function. Additionally, we examine the limitations of Lion and identify scenarios where its improvements are small or not statistically significant.

## 1   Introduction

Optimization algorithms, i.e., optimizers, play a fundamental role in training neural networks. There are a large number of handcrafted optimizers, mostly adaptive ones, introduced in recent years [2, 3, 7, 27, 58, 105]. However, Adam [48] with decoupled weight decay [60], also referred to as AdamW, and Adafactor with factorized second moments [86], are still the de facto standard optimizers for training most deep neural networks, especially the recent state-of-the-art language [12, 23, 90], vision [21, 26, 102] and multimodal [75, 84, 100] models.

---

[*]Work done as a student researcher at Google Brain.

37th Conference on Neural Information Processing Systems (NeurIPS 2023).

Table 1: Accuracy of BASIC-L [72] on ImageNet and several robustness benchmarks. We apply Lion to both vision tower pre-training and vision-language contrastive training stages. The previous SOTA results on *zero-shot* and *fine-tuning* ImageNet accuracy are 86.3% and 91.0% [100].

| Optimizer | Zero-shot | | | | | | Fine-tune |
| | ImageNet | V2 | A | R | Sketch | ObjectNet | ImageNet |
|---|---|---|---|---|---|---|---|
| Adafactor | 85.7 | 80.6 | 85.6 | 95.7 | 76.1 | 82.3 | 90.9 |
| Lion | **88.3** | **81.2** | **86.4** | **96.8** | **77.2** | **82.9** | **91.1** |

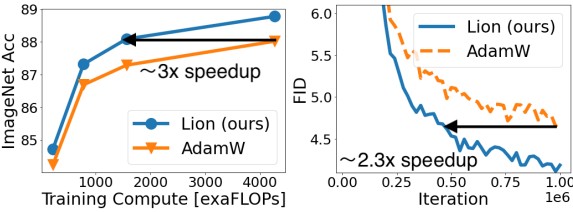

Figure 1: **Left**: ImageNet fine-tuning accuracy vs. pre-training cost of ViT models on JFT-300M. **Right**: FID of the diffusion model on $256^2$ image generation. We use DDPM for 1K steps w/o guidance to decode image. As a reference, the FID of ADM is 10.94 [24].

Program 1: Discovered optimizer Lion. $\beta_1 = 0.9$ and $\beta_2 = 0.99$ by default are derived from Program 4. It only tracks momentum and uses the sign operation to compute the update. The two gray lines compute the standard decoupled weight decay, where $\lambda$ is the strength.

```python
def train(weight, gradient, momentum, lr):
    update = interp(gradient, momentum, β₁)
    update = sign(update)
    momentum = interp(gradient, momentum, β₂)
    weight_decay = weight * λ
    update = update + weight_decay
    update = update * lr
    return update, momentum
```

Another direction is to automatically discover such optimization algorithms. The learning to optimize (L2O) approach proposes to discover optimizers by training parameterized models, e.g., neural networks, to output the updates [1, 54, 63, 64]. However, those black-box optimizers, typically trained on a limited number of small tasks, struggle to generalize to state-of-the-art settings where much larger models are trained with significantly more training steps. Another line of methods [5, 95] apply reinforcement learning or Monte Carlo Sampling to discover new optimizers, where the search space is defined by trees composed from predefined operands (e.g., gradient and momentum) and operators (e.g., unary and binary math operations). However, to make the search manageable, they often limit the search space by using fixed operands and restricting the size of the tree, thereby limiting the potential for discovery. For example, they are *unable* to modify the tracking of momentum or how it contributes to the update, which is an essential component of Lion. Consequently, the algorithms discovered have not yet reached the state-of-the-art. AutoML-Zero [80] is an ambitious effort that attempts to search every component of a machine learning pipeline while evaluating on toy tasks. This work follows the research direction of automatic discovering optimizers and is in particular inspired by AutoML-Zero, but aims at discovering effective optimization algorithms that can improve the state-of-the-art benchmarks.

In this paper, we present a method to formulate algorithm discovery as program search and apply it to discover optimization algorithms. There are two primary challenges. The first one is to find high-quality algorithms in the infinite and sparse program space. The second one is to further select out the algorithms that can generalize from small proxy tasks to much larger, state-of-the-art tasks. To tackle these challenges, we employ a range of techniques including evolutionary search with warm-start and restart, abstract execution, funnel selection, and program simplification.

Our method discovers a simple and effective optimization algorithm: Lion. This optimizer differs from various adaptive algorithms by only tracking momentum and leveraging the sign operation to calculate updates, leading to lower memory overhead and uniform update magnitudes across all dimensions. Despite its simplicity, Lion demonstrates outstanding performance across a range of models (Transformer, MLP, ResNet, U-Net, and Hybrid) and tasks (image classification, vision-language contrastive learning, diffusion, language modeling, and fine-tuning). Notably, we achieve 88.3% *zero-shot* and 91.1% *fine-tuning* accuracy on ImageNet by replacing Adafactor with Lion in BASIC [72], surpassing the previous best results by 2% and 0.1%, respectively. Additionally, Lion reduces the pre-training compute on JFT by up to 5x, improves training efficiency on diffusion models by 2.3x and achieves a better FID score, and offers similar or better performance on language modeling with up to 2x compute savings. Limitations of our work are discussed in Appendix N.

Program 2: An example training loop, where the optimization algorithm that we are searching for is encoded within the `train` function. The main inputs are the weight (`w`), gradient (`g`) and learning rate schedule (`lr`). The main output is the `update` to the weight. `v1` and `v2` are two additional variables for collecting historical information.

```
w = weight_initialize()
v1 = zero_initialize()
v2 = zero_initialize()
for i in range(num_train_steps):
  lr = learning_rate_schedule(i)
  g = compute_gradient(w, get_batch(i))
  update, v1, v2 = train(w, g, v1, v2, lr)
  w = w - update
```

Program 3: Initial program (AdamW). The bias correction and $\epsilon$ are omitted for simplicity.

```
def train(w, g, m, v, lr):
  g2 = square(g)
  m = interp(g, m, 0.9)
  v = interp(g2, v, 0.999)
  sqrt_v = sqrt(v)
  update = m / sqrt_v
  wd = w * 0.01
  update = update + wd
  lr = lr * 0.001
  update = update * lr
  return update, m, v
```

Program 4: Discovered program after search, selection and removing redundancies in the raw Program 8 (Appendix). Some variables are renamed for clarity.

```
def train(w, g, m, v, lr):
  g = clip(g, lr)
  g = arcsin(g)
  m = interp(g, v, 0.899)
  m2 = m * m
  v = interp(g, m, 1.109)
  abs_m = sqrt(m2)
  update = m / abs_m
  wd = w * 0.4602
  update = update + wd
  lr = lr * 0.0002
  m = cosh(update)
  update = update * lr
  return update, m, v
```

## 2 Symbolic Discovery of Algorithms

We present an approach that formulates algorithm discovery as program search [11, 49, 80]. We use a symbolic representation in the form of programs for the following advantages: (1) it aligns with the fact that algorithms must be implemented as programs for execution; (2) symbolic representations like programs are easier to analyze, comprehend and transfer to new tasks compared to parameterized models such as neural networks; (3) program length can be used to estimate the complexity of different programs, making it easier to select the simpler, often more generalizable ones. This work focuses on optimizers for deep neural network training, but the method is generally applicable.

### 2.1 Program Search Space

We adhere to the following three criteria while designing the program search space: (1) the search space should be flexible enough to enable the discovery of novel algorithms; (2) the programs should be easy to analyze and incorporate into a machine learning workflow; (3) the programs should focus on the high-level algorithmic design rather than low-level implementation details. We define the programs to contain functions operating over n-dimensional arrays, including structures like lists and dictionaries containing such arrays, in an imperative language. They are similar to Python code using NumPy / JAX [10, 36] as well as pseudo code of optimization algorithms. The details of the design are outlined below, with an example representation of AdamW in Program 3.

**Input / output signature** The program defines a `train` function, which encodes the optimization algorithm being searched for, where the main inputs are the model weight (`w`), the gradient (`g`) and the learning rate schedule value (`lr`) at the current training step. The main output is the `update` to the weight. The program also incorporates extra variables initialized as zeros to collect historical information during training. For example, AdamW requires two extra variables to estimate first and second moments. Note that those variables can be used arbitrarily, we use the name `m` and `v` in Program 3 just for better readability. This simplified code snippet in Program 2 uses the same signature as AdamW to ensure that the discovered algorithms have smaller or equal memory footprints. As opposed to previous optimizer search attempts [5, 95], our method allows discovering better ways of updating the extra variables.

**Building blocks** The `train` function consists of a sequence of assignment statements, with no restrictions on the number of statements or local variables. Each statement calls a function using constants or existing variables as inputs, and the resulting value is stored in a new or existing variable. For the program, we select 45 common math functions, most of which corresponds to a function in NumPy or an operation in linear algebra. Some functions are introduced to make the program more compact, such as the linear interpolation function `interp(x, y, a)`, which is made equivalent to `(1 - a) * x + a * y`. Preliminary experiments have investigated the inclusion of more advanced features such as conditional and loop statements, and defining and calling new functions, but these

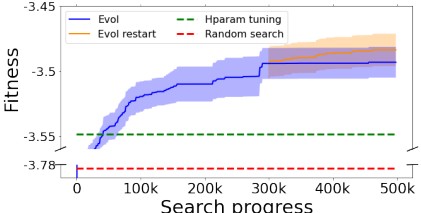 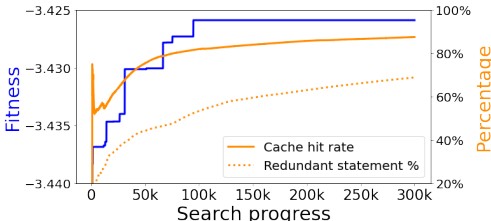

Figure 2: **Left**: We run hyperparameter tuning on AdamW and random search, both with 4x more compute, to get the best results as two baselines (green and red lines). The evolutionary search, with mean and standard error calculated from five runs, significantly outperforms both of them. The use of multiple restarts from the initial program is crucial due to the high variance in the search fitness (blue curves), and restarting from the best program after 300K progress further improves the fitness (orange curves) when the original search plateaus. **Right**: Example curves of search fitness, the cache hit rate, and the percentage of redundant statements. The cache hit rate and the redundant statements percentage increase along with the search progress to $\sim$90% and $\sim$70%.

do not yield improved results, so we leave them out. A detailed description of the functions are summarized in Appendix H. When necessary, the types and shapes of the function arguments are automatically cast, e.g., in the case of adding a dictionary of arrays to a scalar.

**Mutations and redundant statements** The design of mutations utilized in evolutionary search is tightly intertwined with the representation of the program. We include three types of mutations: (1) inserting a new statement at a random location with randomly chosen functions and arguments, (2) deleting a random chosen statement, and (3) modifying a random statement by randomly altering one of its function arguments, which may be either variables or constants. To mutate an argument, we replace it with an existing variable or a newly generated constant obtained by sampling from a normal distribution $X \sim \mathcal{N}(0\ 1)$. Additionally, we can mutate an existing constant by multiplying it by a random factor $2^a$, where $a \sim \mathcal{N}(0\ 1)$. These constants serve as tunable hyperparameters in the optimization algorithm, such as the peak learning rate and weight decay in AdamW. The modification mutation makes it easier for the search to tune those constants while keeping most of the program unchanged. Note that we allow a program to include redundant statements during search, i.e., statements that do not impact the final program outputs. This is necessary as mutations are limited to only affecting a single statement. Redundant statements therefore serve as intermediate steps towards bigger changes.

**Infinite and sparse search space** Given the limitless number of statements and local variables, as well as the presence of mutable constants, the program search space is infinite. Even if we ignore the constants and bound the program length and number of variables, the number of potential programs is still intractably large. More importantly, the challenge comes from the sparsity of high-performing programs in the search space. To illustrate this point, we evaluates 2M randomly sampled programs on a low-cost proxy task. The best program among them is still significantly inferior to AdamW.

## 2.2 Efficient Search Techniques

We employ the following techniques to address the challenges posed by the infinite and sparse space.

**Evolution with warm-start and restart** We apply regularized evolution as it is simple, scalable, and has shown success on many AutoML search tasks [42, 79, 80, 87, 99]. It keeps a population of $P$ algorithms that are gradually improved through cycles. Each cycle picks $T < P$ algorithms at random and the best performer is chosen as the *parent*, i.e., *tournament selection* [32]. This parent is then copied and *mutated* to produce a *child* algorithm, which is added to the population, while the oldest algorithm is removed. Normally, evolutionary search starts with random candidates, but we warm-start the initial population as AdamW to accelerate the search. By default, we use a tournament size of two and a population size of 1K. To further improve the search efficiency, we apply two types of restart: (1) restarting from the initial program, which can lead to different local optima due to the randomness in evolution and encourage exploration. This can be done by running multiple searches in parallel. (2) restarting from the best algorithm found thus far to further optimize it, encouraging exploitation. Figure 2 (Left) displays the mean and standard error of five evolutionary search experiments. We run hyperparameter tuning based on AdamW by only allowing mutations of constants in the evolution, and run random search by sampling random programs, both with 4x more

compute. Our search significantly outperforms the best results from both baselines, shown as the two dashed lines. The high variance in the search fitness necessitates running multiple repeats by restarting from the initial program. When the search fitness plateaus after $\sim$300K progress, restarting from the best program found thus far further improves the fitness shown by the orange curve.

**Pruning through abstract execution** We propose to prune the redundancies in the program space from three sources: programs with syntax or type / shape errors, functionally equivalent programs, and redundant statements in the programs. Before a program is actually executed, we perform an abstract execution step that (1) infers variable types and shapes to detect programs with errors; (2) produces a hash that uniquely identifies how the outputs are computed from the inputs, allowing us to cache and look up semantically duplicate programs [31]; (3) identifies redundant statements that can be ignored during actual execution and analysis. For instance, Program 4 is obtained after removing all redundant statements in Program 8 (Appendix). Abstract execution has negligible cost compared to the actual execution, with each input and function replaced by customized values, e.g., hash. See Appendix I for details of abstract execution. Preliminary experiments have shown that the search process can become overwhelmed with invalid programs and cannot make progress without filtering out invalid programs. As seen in Figure 2 (Right), the percentage of redundant statements and cache hit rate both increase as the search proceeds. Based on five search runs, each covering 300K programs, there are $69.8 \pm 1.9\%$ redundant statements towards the end, implying that redundant statements removal makes the program $\sim$3x shorter on average. The cache hit rate is $89.1 \pm 0.6\%$, indicating that using the hash table as cache brings $\sim$10x reduction on the search cost.

**Proxy tasks and search cost** To reduce search cost, we create low-cost proxies by decreasing the model size, number of training examples, and steps from the target tasks. Evaluation on the proxies can be completed on one TPU V2 chip within 20min. We use the accuracy or perplexity on the validation set as the fitness. Each search experiment utilizes 100 TPU V2 chips and runs for $\sim$72h. There are a total of 200-300K programs generated during each search experiment. However, the number of programs that are actually evaluated is around 20-30K, thanks to the use of the cache through abstract execution. To incorporate restart, we start five repeats of search experiments, followed by another round of search initializing from the best algorithm found thus far. This results in a total cost of $\sim$3K TPU V2 days. See Appendix F for the details of proxy tasks.

## 2.3 Generalization: Program Selection and Simplification

The search experiments can discover promising programs on proxy tasks. We use performance on *meta-validation* tasks that are larger than the proxy tasks by increasing the model size and training steps, to select the programs that generalize beyond proxy tasks then further simplify them. The phenomenon of *meta-overfitting* occurs when the search fitness keeps growing, but the meta-validation metric declines, indicating that the discovered algorithms have overfit the proxy tasks. Two examples meta-validation curves are shown in Figure 11 (Left) in the Appendix.

**Large generalization gap** The discovered algorithms face a significant challenge due to the substantial gap between the proxy tasks during search and the target tasks. While proxy tasks can typically be completed within 20min on one TPU V2 chip, target tasks can be $> 10^4$x larger and require days of training on 512 TPU V4 chips. Furthermore, we expect the optimizer to perform well on different architectures, datasets and even different domains, so the discovered algorithms need to show strong out-of-distribution generalization. The sparse search space and inherent noise in the evolution process further compound this challenge, leading to inconsistent generalization properties between different runs. Our observation suggests that evolutionary search runs that meta-overfit later tend to uncover optimization algorithms that generalize better. See more details in Figure 11 (Right) in the Appendix.

**Funnel selection** To mitigate the generalization gap, we collect promising programs based on search fitness and add an extra selection step using a series of meta-validation tasks to select those generalize better. To save compute, we apply a funnel selection process that gradually increases the scale of the meta-validation tasks. For example, starting with proxy task A, we create a 10x larger task B by increasing the model size and the training steps. Only algorithms that surpass the baseline on task B will be evaluated on task C, which is 100x larger. This approach allows us to gradually filter out algorithms that show poor generalization performance, ultimately leading to the selection of algorithms that generalize well to larger tasks.

**Simplification** Simpler programs are easier to understand and our intuition is that they are more likely to generalize, so we simplify the programs with the following steps. Firstly, we remove redundant statements that do not contribute to the final output as identified through abstract execution. Secondly, we remove statements that are non-redundant but produce minimal differences when removed. This step can also be achieved through evolution by disabling the insertion of new statements in the mutation process. Finally, we rearrange the statements manually, assign clear and descriptive names to variables, and convert the program into its simpler, mathematically equivalent form.

# 3  Derivation and Analysis of Lion

We arrive at the Lion optimizer due to its simplicity, memory efficiency, and strong performance in search and meta-validation. The search also discovers other algorithms shown in Appendix D, e.g., some with better regularization and some resembling AdaBelief [105] and AdaGrad [29].

## 3.1  Derivation

The search and funnel selection process lead to Program 4, which is obtained by automatically removing redundant statements from the raw Program 8 (in the Appendix). We further simplify it to get the final algorithm (Lion) in Program 1. Several unnecessary elements are removed from Program 4 during the simplification process. The `cosh` function is removed since `m` would be reassigned in the next iteration (line 3). The statements using `arcsin` and `clip` are also removed as we observe no quality drop without them. The three red statements translate to a single `sign` function. Although both `m` and `v` are utilized in Program 4, `v` only changes how the momentum is updated (two `interp` functions with constants ∼0.9 and ∼1.1 is equivalent to one with ∼0.99) and does not need to be separately tracked. Note that the bias correction is no longer needed, as it does not change the direction. Algorithm 2 in the Appendix shows the pseudocode.

## 3.2  Analysis

**Sign update and regularization** The Lion algorithm produces update with uniform magnitude across all dimensions by taking the sign operation, which is in principle different from various adaptive optimizers. Intuitively, the sign operation adds noise to the updates, which acts as a form of regularization and helps with generalization [16, 30, 67]. An evidence is shown in Figure 9 (Right) in the Appendix, where the ViT-B/16 trained by Lion on ImageNet has a higher training error compared to AdamW but a 2% higher accuracy on the validation set (as shown in Table 8 from the Appendix). Additionally, the results in Appendix G demonstrate that Lion leads to the convergence in smoother regions, which usually results in better generalization.

**Momentum tracking** The default EMA factor used to track the momentum in Lion is 0.99 ($\beta_2$), compared to the commonly used 0.9 in AdamW and momentum SGD. The current gradient and momentum are interpolated with a factor of 0.9 ($\beta_1$) before the sign operation is applied. This choice of EMA factor and interpolation allows Lion to balance between remembering a ∼10x longer history of the gradient in momentum and putting more weight on the current gradient in the update. The necessity of both $\beta_1$ and $\beta_2$ is further discussed in Appendix L.

**Hyperparameter and batch size choices** Lion is simpler and has fewer hyperparameters compared to AdamW and Adafactor as it does not require $\epsilon$ and factorization-related ones. The update is an element-wise binary ±1 if we omit the weight decay term, with larger norm than those produced by other optimizers like SGD and adaptive algorithms. As a result, Lion needs a *smaller* learning rate and in turn a *larger* decoupled weight decay to achieve a similar effective weight decay strength (lr * $\lambda$). Detailed information on tuning Lion can be found in Appendix M. Additionally, the advantage of Lion over AdamW enlarges as the batch size increases, which fits the common practice of scaling up model training through data parallelism (Appendix L).

**Memory and runtime benefits** Lion only saves the momentum thus has smaller memory footprint than popular adaptive optimizers like AdamW, which is beneficial when training large models and / or using a large batch size. As an example, AdamW needs at least 16 TPU V4 chips to train a ViT-B/16 with image resolution 224 and batch size 4,096, while Lion only needs 8 (both with `bfloat16` momentum). Another practical benefit is that Lion has faster runtime (steps / sec) in our experiments

due to its simplicity, usually 2-15% speedup compared to AdamW and Adafactor depending on the task, codebase, and hardware.

**Relation to existing optimizers** The sign operation has been explored in previous optimizers [7, 83]. The closest to ours is the handcrafted optimizer signSGD [7] (and its momentum variant) that also utilizes the sign operation to calculate the update but has a different momentum update rule from Lion. Their focus is to mitigate communication costs between agents in distributed training, and they observe inferior performance when training ConvNets on image classification tasks. On the other hand, NAdam [27] combines the updated first moment and the gradient to compute the update, but Lion decouples the momentum tracking and how it is applied to the update through $\beta_2$. A comparison of Lion with related optimizers can be found in Appendix K.

## 4 Evaluation of Lion

In this section, we present evaluations of Lion, on various benchmarks. We mainly compare it to AdamW (or Adafactor when memory is a bottleneck) as it is exceedingly popular and the de facto standard optimizer on a majority of learning tasks. The result of momentum SGD is only included for ResNet since it performs worse than AdamW elsewhere. We also benchmark other popular optimizers in Appendix K, including handcrafted and automatically discovered ones. We make sure that every optimizer is well-tuned for each task (see Appendix M for tuning details). By default, the learning rate schedule is cosine decay with 10K steps warmup, and the momentum is saved as `bfloat16` to reduce the memory footprint. Ablations studies are performed in Appendix L.

### 4.1 Image Classification

We perform experiments including various datasets and architectures on the image classification task (see Appendix B for dataset details). Apart from training from scratch on ImageNet, we also pre-train on two larger well-established datasets, ImageNet-21K and JFT [89]. The image size is $224^2$ by default otherwise specified by the subscript.

**Train from scratch on ImageNet** Following previous works [26, 37], we train ResNet-50 for 90 epochs with a batch size of 1,024, and other models for 300 epochs with a batch size of 4,096. As shown in Table 8 (in the Appendix), Lion significantly outperforms AdamW on various architectures. Empirically, the improvement is more substantial on models with larger capacity, with accuracy increases of 1.96% and 0.58% for ViT-B/16 and ViT-S/16, respectively. The performance gaps also tend to enlarger with fewer inductive biases. When strong augmentations are applied, the gain of Lion over AdamW shrinks, but it still outperforms AdamW by 0.42% on CoAtNet-3, despite the strong regularization during training [21].

**Pre-train on ImageNet-21K** We pre-train ViT-B/16 and ViT-L/16 on ImageNet-21K for 90 epochs with a batch size of 4,096. Table 8 shows that Lion surpasses AdamW when the training set is enlarged for 10x. The gaps on larger models are bigger, with +0.52% vs. +0.33% (ImageNet), +0.57% vs. +0.23% (ReaL), and +0.74% vs. +0.25% (V2) for ViT-L/16 and ViT-B/16, respectively.

**Pre-train on JFT** To push the limit, we conduct extensive experiments on JFT. We follow the settings of Dosovitskiy et al. [26] and Zhai et al. [102] for both pre-training and fine-tuning. Figure 1 (Left) and 3 present the accuracy of three ViT models (ViT-B/16, ViT-L/16, and ViT-H/14) under different pre-training budgets on JFT-300M. Lion enables the ViT-L/16 to match the performance of ViT-H/14 trained by AdamW on ImageNet and ImageNet V2 but with 3x less pre-training cost. On ImageNet ReaL, the compute saving further becomes 5x. Another evidence is that even when a ViT-L/16 is trained by AdamW for 4M steps by Zhai et al. [102], its performance still lags behind the same model trained by Lion for 1M steps.

Table 5 in the Appendix shows the fine-tuning results, with higher resolution and EMA. Our ViT-L/16 matches the previous ViT-H/14 results trained by AdamW, while being 2x smaller. The advantage is larger on more challenging benchmarks, such as +1.33% (V2), +6.08% (A), +5.54% (R) for ViT-L/16. When pretrained on JFT-3B, the ViT-g/14 trained by Lion outperforms the previous ViT-G/14 results [102], with 1.8x fewer parameters. Our ViT-G/14 achieves a 90.71% accuracy on ImageNet.

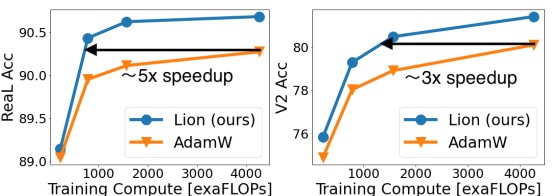

Figure 3: ImageNet ReaL (**Left**) and ImageNet V2 (**Right**) accuracy after we pre-train ViT models on JFT-300M then fine-tune on ImageNet. See Table 4 (in the Appendix) for the detailed numbers.

Table 2: Zero-shot accuracy of LiTs on ImageNet, CIFAR-100, and Oxford-IIIT Pet. As a reference, the zero-shot accuracy of CLIP [75] on ImageNet is 76.2%.

| Model | Optimizer | ImageNet | C100 | Pet |
|---|---|---|---|---|
| LiT-B/32-B | AdamW | 68.78 | 71.41 | 86.62 |
| | Lion | **69.88** | **71.78** | **87.36** |
| LiT-B/16-B | AdamW | 74.26 | 72.25 | 89.83 |
| | Lion | **75.39** | **72.49** | **91.20** |
| LiT-g/14$_{288}$-L | AdamW | 83.43 | 80.93 | 94.88 |
| | Lion | **84.09** | **81.43** | **95.86** |

Table 3: One-shot evaluation averaged over three NLG and 21 NLU tasks. The results of GPT-3 [12] and PaLM [17] are included for reference. The LLMs trained by Lion have better in-context learning ability. See Table 11 (in the Appendix) for detailed results on all tasks.

| Task | 1.1B | | 2.1B | | 7.5B | | 6.7B | 8B |
|---|---|---|---|---|---|---|---|---|
| | Adafactor | Lion | Adafactor | Lion | Adafactor | Lion | GPT-3 | PaLM |
| #Tokens | 300B | | | | | | 300B | 780B |
| Avg NLG | 11.1 | **12.1** | 15.6 | **16.5** | 24.1 | **24.7** | 23.1 | 23.9 |
| Avg NLU | 53.2 | **53.9** | 56.8 | **57.4** | 61.3 | **61.7** | 58.5 | 59.4 |

## 4.2 Vision-Language Contrastive Learning

This section focuses on the vision-language contrastive training [75]. We compare Lion with AdamW (Adafactor) on zero-shot image classification and image-text retrieval benchmarks. We initialize the image encoder with a strong pre-trained model as it is suggested to be more efficient [103].

**Locked-image text Tuning (LiT)** We perform a comparison between Lion and AdamW on LiT [103] by training the text encoder [103] in a contrastive manner using the same frozen pre-trained ViT. All models are trained for 1B image-text pairs with a batch size of 16,384. Table 2 shows the zero-shot image classification results on three model scales, with the name specifies the size, e.g., LiT-B/16-B denotes a ViT-B/16 and a base size Transformer as the text encoder. Our method, Lion, demonstrates consistent improvement over AdamW with gains of +1.10%, +1.13%, and +0.66% on zero-shot ImageNet accuracy for LiT-B/32-B, LiT-B/16-B, and LiT-g/14$_{288}$-L, respectively. Figure 7 (Left) in the Appendix depicts an example zero-shot learning curve of LiT-B/16-B. Similar results are obtained on the other two datasets. The zero-shot image-text retrieval results on MSCOCO [56] and Flickr30K [74] can be found in Figure 6 (in the Appendix). The evaluation metric is Recall@K, calculated based on if the ground truth label of the query appears in the top-K retrieved examples. Lion outperforms AdamW on both datasets, with a larger gain in Recall@1 than Recall@10 on Flicker30K, implying more accurate retrieval results: +1.70% vs. +0.60% for image → text and +2.14% vs. +0.20% for text → image.

**BASIC** Pham et al. [72] propose to scale up batch size, dataset, and model size simultaneously, achieving drastic improvements over CLIP. It uses a sophisticated CoAtNet [21] pre-trained on JFT-5B as the image encoder. Furthermore, the contrastive training is performed on 6.6B image-text pairs with a larger 65,536 batch size. To push the limit, we only experiment on the largest BASIC-L, and use Lion on *both* image encoder pre-training and contrastive learning stages. As illustrated in Table 1, we achieve a significant 2.6% gain over the baseline, striking a 88.3% accuracy on zero-shot ImageNet classification. Note that this result is 2.0% higher than the previous best result [100]. The performance gain is consistent on five other robustness benchmarks. After fine-tuning the image encoder (CoAtNet-7) in BASIC-L obtained by Lion, we further achieve a 91.1% top-1 accuracy on ImageNet, which is 0.1% better than the previous SOTA.

## 4.3 Diffusion Model

Recently, diffusion models achieve a huge success on image generation [24, 40, 41, 84, 88], so we test the performance of Lion on unconditional image synthesis and multimodal text-to-image generation.

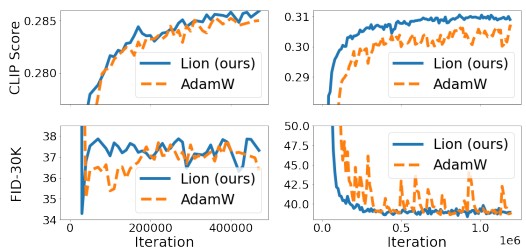

Figure 4: Imagen text-to-image $64^2$ (**Left**) and the $64^2 \rightarrow 256^2$ diffusion models (**Right**).

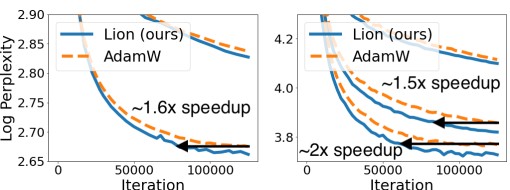

Figure 5: Log perplexity on Wiki-40B (**Left**) and PG-19 (**Right**). The speedup brought by Lion tends to increase with the model scale. The largest model on Wiki-40B is omitted as we observe severe overfitting.

**Image synthesis on ImageNet** We utilize the improved U-Net architecture [24] and perform $64 \times 64$, $128 \times 128$, and $256 \times 256$ image generation on ImageNet. The batch size is 2,048 and the learning rate remains constant throughout training. For decoding, we apply DDPM [41] for 1K sampling steps *without* classifier-free guidance. The evaluation metric is the standard FID score. Illustrated by Figure 1 (Right) and Figure 7 (Middle and Right) in the Appendix, Lion enables both better quality and faster convergence on the FID score. The gap between Lion and AdamW increases with the image resolution, where the generation becomes more challenging. When generating $256 \times 256$ images, Lion achieves the final performance of AdamW with 2.3x fewer steps. The final FID scores are 4.1 (Lion) vs. 4.7 (AdamW). For reference, the FID of ADM [24] is 10.94.

**Text-to-image generation** We follow the Imagen [84] setup to train a base $64 \times 64$ text-to-image model and a $64 \times 64 \rightarrow 256 \times 256$ super-resolution model. All models are trained on a high-quality internal image-text dataset with a batch size of 2,048 and a constant learning rate. Due to computational constraints, our base U-Net has a width of 192 compared to 512 in the original 2B model, while the 600M super-resolution model is identical to the original Imagen setup. Along with the training, 2K images are sampled from the MSCOCO [56] validation set for real-time evaluation. We use the CLIP score to measure image-text alignment and the zero-shot FID-30K to measure image fidelity. Classifier-free guidance [40] with a weight of 5.0 is applied as it has been shown to improve image-text alignment. Figure 4 depicts the learning curve. While there is no clear improvement on the base $64 \times 64$ model, Lion outperforms AdamW on the text-conditional super-resolution model. It achieves a higher CLIP score and has a less noisy FID metric compared to AdamW.

### 4.4 Language Modeling and Fine-tuning

This section focuses on language modeling and fine-tuning. On language-only tasks, we find that tuning $\beta_1$ and $\beta_2$ can improve the quality for both AdamW and Lion. See Appendix M for tuning details. The masked language modeling and fine-tuning results are shown in Appendix J.

**Autoregressive language modeling** We first experiment on two smaller-scale academic datasets Wiki-40B [34] and PG-19 [76]. The employed Transformer spans three scales: small (110M), medium (336M), and large (731M). The architecture details can be found in Appendix E. All models are trained with $2^{18}$ tokens per batch for 125K steps, with a learning rate schedule of 10K steps warmup followed by linear decay. The context length is set to 512 for Wiki-40B and 1,024 for PG-19. Figure 5 illustrates the token-level perplexity for Wiki-40B and word-level perplexity for PG-19. Lion consistently achieves lower validation perplexity than AdamW. It achieves 1.6x and 1.5x speedup when training the medium size model on Wiki-40B and PG-19, respectively. When the model is increased to the large size, the speedup on PG-19 further increases to 2x.

We then conduct larger experiments with the pre-training dataset similar to GLaM [28]. Following GPT-3 [12], we train three models, from 1.1B to 7.5B parameters, on 300B tokens with a batch size of 3M tokens and a context length of 1K. We evaluate them on three natural language generative (NLG) and 21 natural language understanding (NLU) tasks (see Appendix C for details). We observe no difference in perplexity throughout training. Nevertheless, Lion outperforms Adafactor on the average in-context learning ability, as shown in Table 3. Our 7.5B baseline model, trained for 300B tokens, outperforms the 8B PaLM, trained for 780B tokens, demonstrating the strength of our setup. Lion outperforms Adafactor on both NLG and NLU tasks, particularly on the NLG tasks, with an exact match improvement of +1.0, +0.9, and +0.6 for the 1.1B, 2.1B, and 7.5B models, respectively.

## 5  Related Work

Our work lies in the area of AutoML and meta-learning that includes learning to learn [1, 5, 63, 64, 78, 96, 97], neural architecture search [14, 15, 57, 71, 79, 87, 93, 94, 98, 106] and hyperparameter optimization [25, 43, 44, 55], etc. There is also a long history of using evolutionary methods to search for programs, i.e., genetic programming [11, 42, 49]. Our approach builds upon a symbolic search space similar to AutoML-Zero [70, 80]. However, instead of discovering programs with fixed dimensional matrices, vector, and scalars for toy tasks, our goal is to develop programs that operate on n-dimensional arrays and can generalize to state-of-the-art tasks. Other related works include numerous handcrafted optimizers [2, 7, 27, 29, 35, 48, 58, 61, 82, 83, 86, 105].

## 6  Conclusion

This paper aims at discovering optimization algorithms via program search. Despite the challenges from an infinite and sparse space, and large generalization gap between the proxy and target tasks, our method discovers a simple and effective optimizer, Lion, that is memory-efficient and achieves strong generalization across architectures, datasets and tasks.

## Acknowledgements

We would like to thank (in alphabetical order) Angel Yu, Boqing Gong, Chen Cheng, Chitwan Saharia, Daiyi Peng, David So, Hanxiao Liu, Hanzhao Lin, Jeff Lund, Jiahui Yu, Jingru Xu, Julian Grady, Junyang Shen, Kevin Regan, Li Sheng, Liu Yang, Martin Wicke, Mingxing Tan, Mohammad Norouzi, Qiqi Yan, Rakesh Shivanna, Rohan Anil, Ruiqi Gao, Steve Li, Vlad Feinberg, Wenbo Zhang, William Chan, Xiao Wang, Xiaohua Zhai, Yaguang Li, Yang Li, Zhuoshu Li, Zihang Dai, Zirui Wang for helpful discussions, and the Google Brain team at large for providing a supportive research environment.

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

## Acknowledgments and Disclosure of Funding

Use unnumbered first level headings for the acknowledgments. All acknowledgments go at the end of the paper before the list of references. Moreover, you are required to declare funding (financial activities supporting the submitted work) and competing interests (related financial activities outside the submitted work). More information about this disclosure can be found at: `https://neurips.cc/Conferences/2023/PaperInformation/FundingDisclosure`.

Do **not** include this section in the anonymized submission, only in the final paper. You can use the `ack` environment provided in the style file to autmoatically hide this section in the anonymized submission.

Table 4: Model performance when pre-trained on JFT-300M then fine-tuned on ImageNet. Those numbers correspond to Figure 1 (Left) and Figure 3. The fine-tuning resolution is $384^2$ for ViT-B/16 and ViT-L/16, and $392^2$ for ViT-H/14. Following Dosovitskiy et al. [26], Polyak averaging is not applied here.

| Model | #Params | Epochs / Steps | Optimizer | ImageNet | ReaL | V2 | A | R |
|---|---|---|---|---|---|---|---|---|
| ViT-B/16$_{384}$ | 86.86M | 7 / 517,791 | AdamW | 84.24 | 89.04 | 74.89 | 27.39 | 53.71 |
| | | | Lion | **84.72** | **89.14** | **75.83** | **29.65** | **55.86** |
| ViT-L/16$_{384}$ | 304.72M | 7 / 517,791 | AdamW | 86.69 | 89.95 | 78.03 | 40.55 | 64.47 |
| | | | Lion | **87.32** | **90.43** | **79.29** | **47.13** | **68.49** |
| | | 14 / 1,035,583 | AdamW | 87.29 | 90.11 | 78.91 | 42.56 | 64.34 |
| | | | Lion | **88.09** | **90.62** | **80.48** | **51.55** | **70.72** |
| ViT-H/14$_{392}$ | 632.72M | 14 / 1,035,583 | AdamW | 88.02 | 90.27 | 80.10 | 53.14 | 69.48 |
| | | | Lion | **88.78** | **90.68** | **81.41** | **58.21** | **73.09** |

Table 5: Model performance when pre-trained on JFT then fine-tuned on ImageNet. Two giant ViT models are pre-trained on JFT-3B while smaller ones are pre-trained on JFT-300M. The ViT-G/14 results are directly from Zhai et al. [102].

| Model | ViT-L/16$_{512}$ | | ViT-H/14$_{518}$ | | ViT-g/14$_{518}$ | | ViT-G/14$_{518}$ | |
|---|---|---|---|---|---|---|---|---|
| #Params | 305.18M | | 633.47M | | 1.04B | | 1.88B | |
| Optimizer | AdamW | Lion | AdamW | Lion | Adafactor | Lion | Adafactor | Lion |
| ImageNet | 87.72 | **88.50** | 88.55 | **89.09** | 90.25 | **90.52** | 90.45 | **90.71 / 90.71**[*] |
| ReaL | 90.46 | **90.91** | 90.62 | **91.02** | 90.84 | **91.11** | 90.81 | **91.06 / 91.25**[*] |
| V2 | 79.80 | **81.13** | 81.12 | **82.24** | 83.10 | **83.39** | 83.33 | **83.54 / 83.83**[*] |
| A | 52.72 | **58.80** | 60.64 | **63.78** | - | - | - | - |
| R | 66.95 | **72.49** | 72.30 | **75.07** | - | - | - | - |

[*] We observe overfitting in fine-tuning, therefore report both the last and oracle results.

# A Pseudocode for AdamW and Lion

---
**Algorithm 1** AdamW Optimizer
---

**given** $\beta_1, \beta_2, \epsilon, \lambda, \eta, f$
**initialize** $\theta_0, m_0 \leftarrow 0, v_0 \leftarrow 0, t \leftarrow 0$
**while** $\theta_t$ not converged **do**
    $t \leftarrow t + 1$
    $g_t \leftarrow \nabla_\theta f(\theta_{t-1})$
    **update EMA of** $g_t$ **and** $g_t^2$
    $m_t \leftarrow \beta_1 m_{t-1} + (1 - \beta_1)g_t$
    $v_t \leftarrow \beta_2 v_{t-1} + (1 - \beta_2)g_t^2$
    **bias correction**
    $\hat{m}_t \leftarrow m_t/(1 - \beta_1^t)$
    $\hat{v}_t \leftarrow v_t/(1 - \beta_2^t)$
    **update model parameters**
    $\theta_t \leftarrow \theta_{t-1} - \eta_t(\hat{m}_t/(\sqrt{\hat{v}_t} + \epsilon) + \lambda\theta_{t-1})$
**end while**
**return** $\theta_t$

---
**Algorithm 2** Lion Optimizer (ours)
---

  **given** $\beta_1, \beta_2, \lambda, \eta, f$
  **initialize** $\theta_0, m_0 \leftarrow 0$
  **while** $\theta_t$ not converged **do**
      $g_t \leftarrow \nabla_\theta f(\theta_{t-1})$
      **update model parameters**
      $c_t \leftarrow \beta_1 m_{t-1} + (1 - \beta_1)g_t$
      $\theta_t \leftarrow \theta_{t-1} - \eta_t(\text{sign}(c_t) + \lambda\theta_{t-1})$
      **update EMA of** $g_t$
      $m_t \leftarrow \beta_2 m_{t-1} + (1 - \beta_2)g_t$
  **end while**
  **return** $\theta_t$

# B Image Classification Tasks

Our evaluation covers various benchmarks: ImageNet, ImageNet ReaL [8], ImageNet V2 [81], ImageNet A [39], ImageNet R [38], ImageNet Sketch [92], ObjectNet [4], CIFAR-100 [50], and Oxford-IIIT Pet [69].

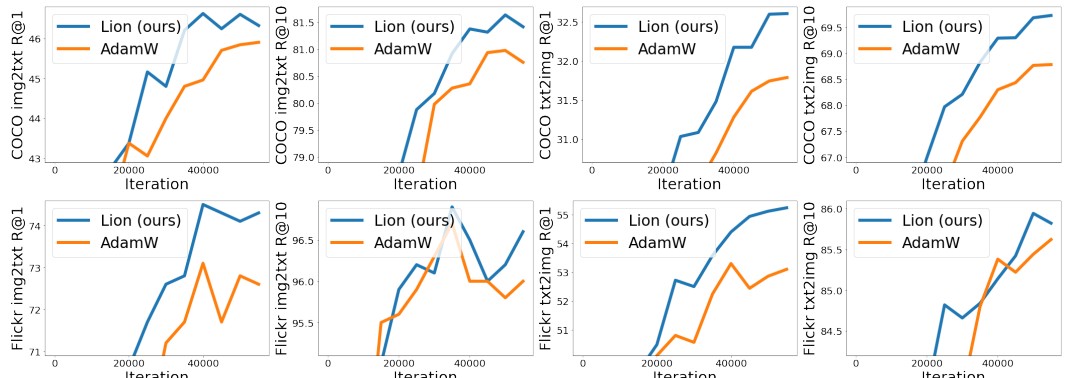

Figure 6: Zero-shot image-text retrieval results on MSCOCO (**Top**) and Flickr30K (**Bottom**) for LiT-B/16-B. Recall@K is calculated based on if the ground truth label of the query appears in the top-K retrieved examples.

## C  NLP Tasks

This section shows all the natural language generation (NLG) and natural language understanding (NLU) tasks where we evaluate the large-scale language models in Section 4.4. Those tasks include *Open-Domain Question Answering*, *Cloze and Completion Tasks*, *Winograd-Style Tasks*, *Common Sense Reasoning*, *In-Context Reading Comprehension*, *SuperGLUE*, and *Natural Language Inference*.

- NLG: TriviaQA [45], Natural Questions [51], Web Questions [6].
- NLU: HellaSwag [101], StoryCloze [66], Winograd [53], Winogrande [85], RACE [52], PIQA [9], ARC [19], OpenbookQA [65], BoolQ [18], Copa [33], RTE [20], WiC [73], Multirc [47], WSC [53], ReCoRD [104], CB [22], Adversarial NLI [68].

Program 5: Algorithm with a better regularization. It dynamically calculates the dot product between the weight and gradient, before computing the weight decay.

```
def train(w, g, m, v, lr):
 m = interp(m, g, 0.16)
 g2 = square(g)
 v = interpolate(v, g2, 0.001)
 v753 = dot(g, w)
 sqrt_v = sqrt(v)
 update = m / sqrt_v
 wd = v753 * w
 update = sin(update)
 update = update + wd
 lr = lr * 0.0216
 update = update * lr
 v = sin(v)
 return update, m, v
```

Program 6: Algorithm that tracks the second moment without EMA decay, which is the same as AdaGrad.

```
def train(w, g, m, v, lr):
 m = interp(m, g, 0.1)
 g2 = square(g)
 g2 = v + g2
 v = interp(v, g2, 0.0015)
 sqrt_v = sqrt(v)
 update = m / sqrt_v
 v70 = get_pi()
 v = min(v, v70)
 update = sinh(update)
 lr = lr * 0.0606
 update = update * lr
 return update, m, v
```

Program 7: Algorithm uses the difference between gradient and momentum to track the second moment, resembling AdaBelief.

```
def train(w, g, m, v, lr):
 m = interp(m, g, 0.1)
 g = g - m
 g2 = square(g)
 v = interp(v, g2, 0.001)
 sqrt_v = sqrt(v)
 update = m / sqrt_v
 wd = w * 0.0238
 update = update + wd
 lr = lr * 0.03721
 update = update * lr
 return update, m, v
```

## D  Other Discovered Programs

By varying the task setting, different types of algorithms can be discovered. For example, if we reduce the amount of data in the proxy task, we are more likely to discover algorithms with better regularization (Program 5), and if we reduce the search progress, we are likely to find simple variants of AdamW (Program 6 and 7). Future work can explore this potential to discover optimizers specialized for different tasks.

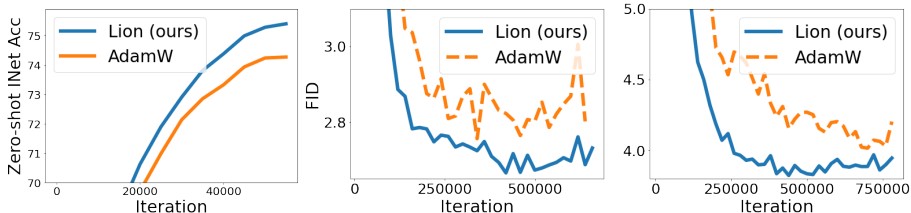

Figure 7: The zero-shot ImageNet accuracy curve of LiT-B/16-B (**Left**). FID comparison on $64 \times 64$ (**Middle**) and $128 \times 128$ (**Right**) image generation when training diffusion models.

Table 6: Architecture details for language modeling.

| Model | #Params | $n_{layers}$ | $d_{model}$ | $n_{heads}$ | $d_{head}$ |
|---|---|---|---|---|---|
| Small-scale | | | | | |
| Small | 110M | 12 | 768 | 12 | 64 |
| Medium | 336M | 24 | 1024 | 16 | 64 |
| Large | 731M | 24 | 1536 | 16 | 96 |
| Large-scale | | | | | |
| 1.1B | 1.07B | 24 | 1536 | 16 | 96 |
| 2.1B | 2.14B | 32 | 2048 | 16 | 128 |
| 7.5B | 7.49B | 32 | 4096 | 32 | 128 |

Table 7: Training error $L_{train}$ and landscape flatness $L_{train}^{\mathcal{N}}$ of ViT-B/16 trained from scratch on ImageNet.

| Optimizer | AdamW | Lion |
|---|---|---|
| ImageNet | 75.48 | 77.44 |
| ReaL | 80.64 | 82.57 |
| V2 | 61.87 | 64.81 |
| $L_{train}$ | 0.61 | 0.75 |
| $L_{train}^{\mathcal{N}}$ | 3.74 | 1.37 |

# E    Architecture Details for Language Modeling

Table 6 shows the Transformer architecture details for language modeling (Section 4.4). The dimension of the feed-forward layer is $4 \times d_{model}$. We use vocabulary size 32K for small-scale and 256K for large-scale models.

# F    Details of Proxy Tasks

For vision tasks, we train a ViT with three layers, 96 hidden units and three heads, on 10% ImageNet for 30k steps with batch size 64. The image size is $64 \times 64$ and the patch size is 16. For language tasks, we train a Transformer with two layers, 128 hidden units and two heads on LM1B [13] for 20K steps with batch size 64, sequence length 32 and vocabulary size 3K. The evaluation time may vary for different programs, but typically a evaluation can be done on one TPU V2 chip within 20min. The validation accuracy or perplexity is used as the fitness.

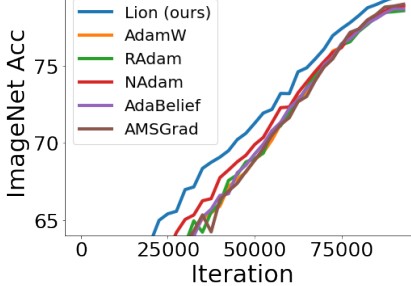
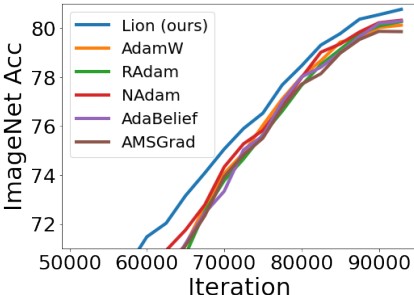

Figure 8: Learning curve of ViT-S/16 (**Left**) and ViT-B/16 (**Right**) associated with Table 10. The curves of the five adaptive optimizers are similar to each other.

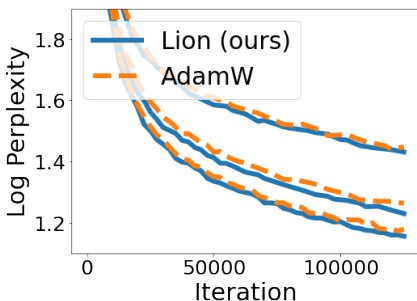
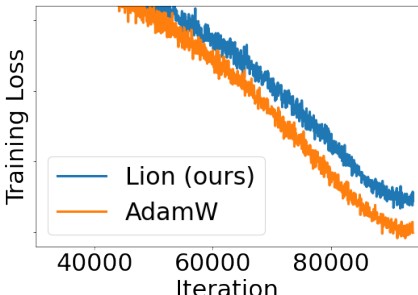

Figure 9: **Left**: Validation perplexity when we perform masked language modeling on the C4 dataset. **Right**: Training loss of ViT-B/16 on ImageNet.

## G  Analysis of Loss Landscape

In this section, we try to understand why our Lion optimizer achieves better generalization than AdamW from the lens of loss geometry. The convergence to a smooth landscape has been shown to benefit the generalization of deep neural networks [14, 16, 30, 46]. Following Chen et al. [16], we measure the landscape flatness at convergence by $L_{train}^{\mathcal{N}} = \mathbb{E}_{\epsilon \sim \mathcal{N}}[L_{train}(w + \epsilon)]$ (average over 1K random noises) in Table 7. We observe that the ViT-B/16 trained by AdamW enjoys a smaller training error $L_{train}$. However, Lion can enable ViT to converge to flatter regions, as it helps the model retain comparably lower error against Gaussian perturbations.

## H  Available Functions

We include 43 available functions that can be used in the program during search. Note that the input of the functions can be one n-dimensional array, dictionaries or lists of arrays, similar to the *pytrees* in JAX.

**Basic math functions from NumPy / JAX** This includes unary functions like abs, cos, sin, tan, arcsin, arccos, arctan, exp, log, sinh, cosh, tanh, arcsinh, arccosh, arctanh, sign, exp2, exp10, expm1, log10, log2, log1p, square, sqrt, cube, cbrt, sign, reciprocal and binary functions like +, -, *, /, power, maximum, minimum with the same semantic as the corresponding function in NumPy / JAX.

**Linear algebra functions commonly used in first-order optimization algorithms** This includes: (1) unary function norm that computes the norm of each arrays in the input; (2) unary function global_norm that computes the global norm by treating all the numbers in the input as one vector; (3) binary function dot that treats the two inputs as two vectors and computes their dot product; (4) binary function cosine_sim that treats the two inputs as two vectors and computes their cosine similarity; (5) binary clip_by_global_norm (clip) that clips the global norm of the first input to the value of the second input that is required to be a scalar; (6) ternary function interpolate (interp) that uses the third argument a, required to be a scalar, to compute a linear interpolation of the first two arguments x and y with (1 - a) * x + a * y.

Program 8: Raw program of Lion before removing redundant statements.

```
def train(w, g, m, v, lr):
  g = clip(g, lr)
  m = clip(m, lr)
  v845 = sqrt(0.6270633339881897)
  v968 = sign(v)
  v968 = v - v
  g = arcsin(g)
  m = interp(g, v, 0.8999999761581421)
  v1 = m * m
  v = interp(g, m, 1.109133005142212)
  v845 = tanh(v845)
  lr = lr * 0.0002171761734643951
  update = m * lr
  v1 = sqrt(v1)
  update = update / v1
  wd = lr * 0.4601978361606598
  v1 = square(v1)
  wd = wd * w
  m = cosh(update)
  lr = tan(1.4572199583053589)
  update = update + wd
  lr = cos(v845)
  return update, m, v
```

**Functions producing commonly used constants** This includes get_pi, get_e, get_eps that generates $\pi$, $e$ and $\epsilon = 10^{-8}$ respectively.

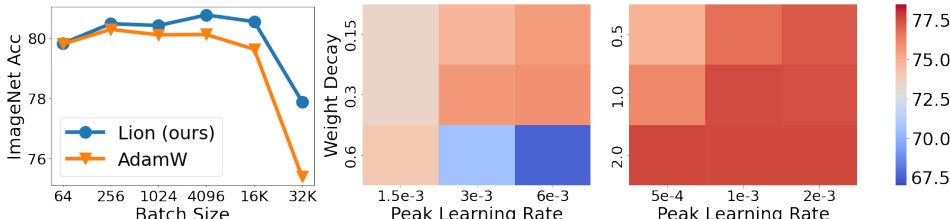

Figure 10: **Left**: Ablation for the effect of batch size. Lion prefers a larger batch than AdamW. ImageNet accuracy of ViT-B/16 trained from scratch when we vary $lr$ and $\lambda$ for AdamW (**Middle**) and Lion (**Right**). Lion is more robust to different hyperparameter choices.

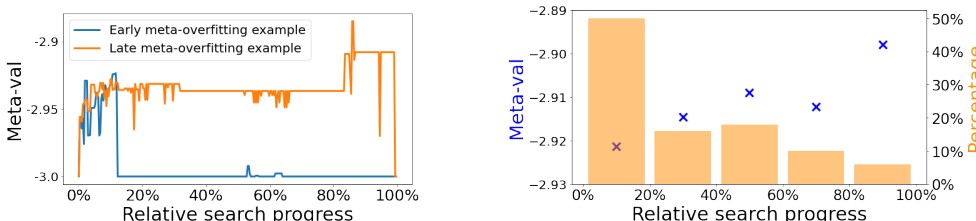

Figure 11: **Left**: The meta-validation (defined in Section 2.3) curves of two search runs measured on a $\sim$500x larger meta-validation task compared to the proxy. The blue one meta-overfits at $\sim$15% of the search progress, while the orange one meta-overfits at $\sim$90% and achieves a better metric. **Right**: Histogram of the search progress when meta-overfitting happens based on 50 runs. Half of the runs meta-overfit early but a long tail of runs meta-overfit much later. Blue cross depicts the best meta-validation metric averaged within each bin, indicating that meta-overfitting happening later leads to programs that generalize better.

# I  Abstract Execution

We propose to prune the large search space with abstract execution. Our approach is motivated by the fact that a large number of programs are invalid, functionally equivalent, or contain redundant statements that waste compute during evaluation. To address this, we introduce an abstract execution step that checks the type and shape of each variable, and computes a hash for each unique computation from inputs to outputs to detect redundant statements. The abstract execution can be seen as a static analysis of the program, achieved by replacing functions and inputs with customized values. We outline the specifics of the customized values and abstract execution procedure for three use cases below. The cost of the abstract execution is usually negligible compared to the actual execution.

**Detecting errors with type / shape inference** To detect programs containing errors, we infer the type and shape of each variable in the program through the following steps: (1) replace each input with an abstract object that only contains type and shape information, and replace each statement with a type and shape inference function; (2) iterate through all statements. Instead of executing the

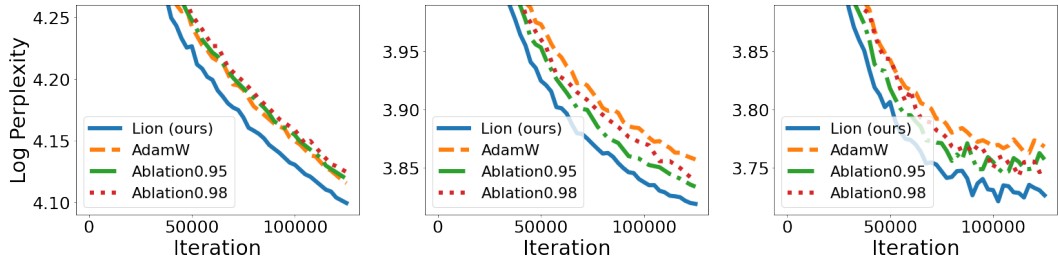

Figure 12: Log perplexity of the small (**Left**), medium (**Middle**), and large (**Right**) size Transformer on PG-19. Since $\beta_1 = 0.95, \beta_2 = 0.98$ in Lion when performing language modeling, we compare to Ablation$_{0.95}$ and Ablation$_{0.98}$ with $\beta = 0.95$ and $\beta = 0.98$, respectively (see Section L for the definition). Lion is still the best-performing one.

Table 8: Accuracy on ImageNet, ImageNet ReaL, and ImageNet V2. Numbers in (·) are from Dai et al. [21], Dosovitskiy et al. [26]. Results are averaged from three runs.

| Model | #Params | Optimizer | RandAug + Mixup | ImageNet | ReaL | V2 |
|---|---|---|---|---|---|---|
| *Train from scratch on ImageNet* | | | | | | |
| ResNet-50 | 25.56M | SGD | ✗ | 76.22 | 82.39 | 63.93 |
| | | AdamW | | 76.34 | **82.72** | **64.24** |
| | | Lion | | **76.45** | **82.72** | 64.02 |
| Mixer-S/16 | 18.53M | AdamW | ✗ | 69.26 | 75.71 | 55.01 |
| | | Lion | | **69.92** | **76.19** | **55.75** |
| Mixer-B/16 | 59.88M | AdamW | ✗ | 68.12 | 73.92 | 53.37 |
| | | Lion | | **70.11** | **76.60** | **55.94** |
| ViT-S/16 | 22.05M | AdamW | ✗ | 76.12 | 81.94 | 63.09 |
| | | Lion | | **76.70** | **82.64** | **64.14** |
| | | AdamW | ✓ | 78.89 | 84.61 | 66.73 |
| | | Lion | | **79.46** | **85.25** | **67.68** |
| ViT-B/16 | 86.57M | AdamW | ✗ | 75.48 | 80.64 | 61.87 |
| | | Lion | | **77.44** | **82.57** | **64.81** |
| | | AdamW | ✓ | 80.12 | 85.46 | 68.14 |
| | | Lion | | **80.77** | **86.15** | **69.19** |
| CoAtNet-1 | 42.23M | AdamW | ✓ | 83.36 (83.3) | - | - |
| | | Lion | | **84.07** | - | - |
| CoAtNet-3 | 166.97M | AdamW | ✓ | 84.45 (84.5) | - | - |
| | | Lion | | **84.87** | - | - |
| *Pre-train on ImageNet-21K then fine-tune on ImageNet* | | | | | | |
| ViT-B/16$_{384}$ | 86.86M | AdamW | ✗ | 84.12 (83.97) | 88.61 (88.35) | 73.81 |
| | | Lion | | **84.45** | **88.84** | **74.06** |
| ViT-L/16$_{384}$ | 304.72M | AdamW | ✗ | 85.07 (85.15) | 88.78 (88.40) | 75.10 |
| | | Lion | | **85.59** | **89.35** | **75.84** |

original statement, we validate a function call by checking the function signature and type and shape information of its arguments. If valid, we compute the type and shape information of the output and assign it to the new variable; (3) verify the validity of the derived type and shape of the output. This process essentially performs a static analysis of the program, exposing errors caused by type and shape mismatch. Note that there are still run-time errors, such as division by zero, that cannot be detected in this manner. Without such filtering of invalid programs, the search would be overwhelmed with invalid programs, making it difficult to achieve meaningful progress.

**Deduplicating with functional hash** Among the valid programs that execute without errors, there are still lots of duplicates due to functionally equivalent programs that have different surface forms but the same underlying functionality. To address this issue, we calculate a functional hash value for every unique computation from the inputs to the outputs as follows: (1) a unique hash value is assigned to each input and function; (2) iterate through all statements, calculating the hash value of the outputs by combining the hash values of the functions and arguments; (3) compute the hash value of program by combining the hash values of all outputs. We then build a hash table that maps each unique functional hash value to the fitness of the corresponding program. When a new program is generated, we first look up its hash value and only perform evaluation if it is not found or if we want to evaluate it multiple times to reduce measurement noise. In our experiments, this technique reduces the search cost by ∼10x, as depicted in Figure 2 (Right).

**Identifying redundant statements by tracking dependencies** In program evolution, redundant statements are included to enable combining multiple mutations to make larger program changes. However, these redundant statements increase the evaluation cost and make program analysis more challenging. To identify redundant statements, we need to determine the set of statements that the outputs depend on, which can be computed in a recursive manner using the following steps: (1)

Table 9: Fine-tuning performance of the T5 Base, Large, and 11B on the GLUE dev set. Results reported are the peak validation scores per task.

| Model | Optimizer | CoLA | SST-2 | MRPC | STS-B | QQP | MNLI -m | MNLI -mm | QNLI | RTE | Avg |
|---|---|---|---|---|---|---|---|---|---|---|---|
| Base | AdamW | 60.87 | 95.18 | 92.39 / 89.22 | **90.70 / 90.51** | 89.23 / 92.00 | 86.77 | 86.91 | 93.70 | 81.59 | 87.42 |
| | Lion | **61.07** | **95.18** | **92.52 / 89.46** | 90.61 / 90.40 | **89.52 / 92.20** | **87.27** | **87.25** | **93.85** | **85.56** | **87.91** |
| Large | AdamW | 63.89 | 96.10 | 93.50 / 90.93 | 91.69 / 91.56 | 90.08 / 92.57 | 89.69 | 89.92 | 94.45 | 89.17 | 89.46 |
| | Lion | **65.12** | **96.22** | **94.06 / 91.67** | **91.79 / 91.60** | **90.23 / 92.67** | **89.85** | **89.94** | **94.89** | **90.25** | **89.86** |
| 11B | AdamW | 69.50 | 97.02 | 93.75 / 91.18 | 92.57 / 92.61 | 90.45 / 92.85 | **92.17** | **91.99** | 96.41 | 92.42 | 91.08 |
| | Lion | **71.31** | **97.13** | **94.58 / 92.65** | **93.04 / 93.04** | **90.57 / 92.95** | 91.88 | 91.65 | **96.56** | **93.86** | **91.60** |

replace the value of each input with an empty set, as they do not depend on any statement; (2) iterate through each statement. Note that each statement is an assignment that calls a function and assigns the result to a variable, which in turn depends on the current statement and all the depending statements of the function arguments. Therefore we replace the value of the variable with its dependency, i.e., a set of all depending statements; (3) compute the union of all statements that each output depends on, which contains all non-redundant statements. By filtering out redundant statements, we obtain a simplified version of the program that is cheaper to execute and easier to analyze. In our experiments, this reduces the program length by ∼3x on average, as shown in Figure 2 (Right).

## J   Masked Language Modeling and Fine-tuning

**Masked language modeling** We also perform BERT training on the C4 dataset [77]. It requires the language models to reconstruct randomly masked out tokens in the input sequence. We use the same architectures and training setups as the smaller-scale autoregressive experiments. Lion performs slightly better than AdamW regarding the validation perplexity: 4.18 vs. 4.25 (small), 3.42 vs. 3.54 (medium), and 3.18 vs. 3.25 (large). See Figure 9 (Left) in the Appendix for the learning curves.

**Fine-tuning** We fine-tune Base (220M), Large (770M), and the largest 11B T5 [77] on the GLUE benchmark [91]. Every model is fine-tuned for 500K steps with a batch size of 128 and a constant learning rate. Table 9 shows the results on the GLUE dev set. For MRPC and QQP, we report the F1 / Accuracy scores, for STS-B, we report the Pearson / Spearman correlation, and for the other datasets, we report their default metric. On average, Lion beats AdamW across all three model scales. It achieves 10, 12, and 10 wins out of 12 scores for T5 Base, Large, and 11B models, respectively.

## K   Comparison with Other Popular Optimizers

We also employ four popular handcrafted optimizers: RAdam [58], NAdam [27], AdaBelief [105], AMSGrad [82] and two optimizers discovered by AutoML: PowerSign [5] and AddSign [5] to train ViT-S/16 and ViT-B/16 on ImageNet (with RandAug and Mixup). We thoroughly tune the peak learning rate $lr$ and decoupled weight decay $\lambda$ [60] of every optimizer, while other hyperparameters are set as the default values in Optax.[2] As shown in Table 10, Lion is still the best performing one. We notice that there is no clear winner amongst the baselines. AMSGrad performs the best on ViT-S/16 but the worst on ViT-B/16. The inferior performance of PowerSign and AddSign compared to other optimizers is consistent with previous observations that automatically discovered optimizers have difficulty generalizing to real-world learning tasks. Figure 8 further shows that the learning curves of the five adaptive optimizers are pretty similar, whereas Lion has a unique one that learns faster.

## L   Ablations

**Momentum tracking** To ablate the effects of both $\beta_1$ and $\beta_2$, we compare to a simple update rule: `m = interp(g, m, ` $\beta$ `); update = sign(m)`. Two optimizers, Ablation$_{0.9}$ and Ablation$_{0.99}$, are created with $\beta$ values of 0.9 and 0.99 respectively. Illustrated by Table 10, the two ablated optimization algorithms perform worse than all five compared baselines, let alone our Lion. Further

---

[2] `https://github.com/deepmind/optax`

Table 10: The performance of various optimizers to train ViT-S/16 and ViT-B/16 on ImageNet (with RandAug and Mixup). Lion is still the best performing one.

| Model | Task | AdamW | RAdam | NAdam | Ada-Belief | AMSGrad | Power-Sign | Add-Sign | Ablation$_{0.9}$ | Ablation$_{0.99}$ | Lion |
|---|---|---|---|---|---|---|---|---|---|---|---|
| ViT-S/16 | ImageNet | 78.89 | 78.59 | 78.91 | 78.71 | 79.01 | 77.36 | 77.37 | 78.23 | 78.19 | **79.46** |
| | ReaL | 84.61 | 84.47 | 84.62 | 84.56 | 85.01 | 83.39 | 83.36 | 84.28 | 84.17 | **85.25** |
| | V2 | 66.73 | 66.39 | 66.02 | 66.35 | 66.82 | 65.17 | 64.52 | 66.13 | 65.96 | **67.68** |
| ViT-B/16 | ImageNet | 80.12 | 80.26 | 80.32 | 80.29 | 79.85 | 78.95 | 78.50 | 79.54 | 79.90 | **80.77** |
| | ReaL | 85.46 | 85.45 | 85.44 | 85.48 | 85.16 | 84.76 | 84.49 | 85.10 | 85.36 | **86.15** |
| | V2 | 68.14 | 67.76 | 68.46 | 68.19 | 68.48 | 67.46 | 65.95 | 68.07 | 68.20 | **69.19** |

ablation studies on the language modeling task (as depicted in Figure 12 in the Appendix) yield similar conclusions. Those results validate the effectiveness and necessity of using two linear interpolation functions, letting Lion to remember longer gradient history meanwhile assign a higher weight to the current gradient.

**Effect of batch size** Some may question whether Lion requires a large batch size to accurately determine the direction due to the added noise from the sign operation. To address this concern, we train a ViT-B/16 model on ImageNet using various batch sizes while maintaining the total training epoch as 300, and incorporating RandAug and Mixup techniques. As shown in Figure 10 (Left), the optimal batch size for AdamW is 256, while for Lion is 4,096. This indicates that Lion indeed prefers a larger batch size, but its performance remains robust even with a small 64 batch size. Furthermore, when the batch size enlarges to 32K, leading to only 11K training steps, Lion achieves a significant 2.5% accuracy gain over AdamW (77.9% vs. 75.4%), demonstrating its effectiveness in the large batch training setting.

# M  Hyperparameter Tuning

To ensure a fair comparison, we tune the peak learning rate $lr$ and decoupled weight decay $\lambda$ for both AdamW (Adafactor) and our Lion using a logarithmic scale. The default values for $\beta_1$ and $\beta_2$ in AdamW are set as 0.9 and 0.999, respectively, with an $\epsilon$ of $1e-8$, while in Lion, the default values for $\beta_1$ and $\beta_2$ are discovered through the program search process and set as 0.9 and 0.99, respectively. We only tune those hyperparameters in Section 4.4, where $\beta_1 = 0.9$, $\beta_2 = 0.99$ in AdamW, and $\beta_1 = 0.95$, $\beta_2 = 0.98$ in Lion. In our experience, reducing $\beta_2$ results in shorter memorization of historical information and *enhanced training stability*. Additionally, the $\epsilon$ in AdamW is set as $1e-6$ instead of the default $1e-8$ as it improves stability in our experiments, similar to the observations in RoBERTa [59].

The update generated by Lion is an element-wise binary $\pm 1$, as a result of the sign operation, therefore it has a larger norm than those generated by other optimizers. Based on our experience, *a suitable learning rate for Lion is typically 3-10x smaller than that for AdamW*. Note that the initial value, peak value, and end value of the learning rate should be changed *simultaneously* with the same ratio compared to AdamW. We *do not* modify other training settings such as the learning rate schedule, gradient and update clipping. Since the effective weight decay is $lr * \lambda$: `update += w * λ`; `update *= lr`, *the value of $\lambda$ used for Lion is 3-10x larger than that for AdamW in order to maintain a similar strength.* For instance,

- $lr = 1e-4$, $\lambda = 10.0$ in Lion and $lr = 1e-3$, $\lambda = 1.0$ in AdamW when training ViT-B/16 on ImageNet with strong augmentations,
- $lr = 3e-5$, $\lambda = 0.1$ in Lion and $lr = 3e-4$, $\lambda = 0.01$ in AdamW for diffusion models,
- $lr = 1e-4$, $\lambda = 0.01$ in Lion and $lr = 1e-3$, $\lambda = 0.001$ in Adafactor for the 7.5B language modeling.

Please see Table 12 (in the Appendix) for all hyperparameters.

Apart from the peak performance, the sensitivity to hyperparameters and the difficulty in tuning them are also critical for the adoption of an optimizer in practice. In Figure 10 (Middle and Right), we alter both $lr$ and $\lambda$ when training ViT-B/16 from scratch on ImageNet. Suggested by the heatmaps, Lion is more robust to different hyperparameter choices compared to AdamW.

# N Limitations

**Limitations of search** Despite the efforts to make the search space less restrictive, it remains inspired by the popular first-order optimization algorithms, leading to a bias towards similar algorithms. It also lacks the functions required to construct advanced second-order algorithms [2, 35, 62]. The search cost is still quite large and the algorithm simplification requires manual intervention. Further reducing the bias in the search space to discover more novel algorithms and improving the search efficiency are important future directions. The current program structure is quite simplistic, as we do not find a good usage of more advanced program constructs such as conditional, loop statements, and defining new functions. Exploring how to incorporate these elements has the potential to unlock new possibilities.

**Limitations of Lion** While we endeavour to evaluate Lion on as many tasks as possible, the assessment is limited to the chosen tasks. On vision tasks, the discrepancies between Lion, AdamW, and momentum SGD are pretty small on ResNets, likely due to the fact that ConvNets are easier to optimize compared to Transformers. The performance gain brought by Lion decreases when strong augmentations are utilized. There are also several tasks where Lion performs similarly to AdamW, including: (1) the Imagen text-to-image base model, (2) the perplexity of autoregressive language model trained on the large-scale internal dataset, which is arguably a more reliable metric the in-context learning benchmarks, and (3) masked language modeling on C4. These tasks have a common characteristic in that the datasets are massive and of high quality, which results in a reduced difference between optimizers. Another potential limitation is the batch size. Though people often scale up the batch size to enable more parallelism, it is likely that Lion performs no better than AdamW if the batch size is small (<64). Additional, Lion still requires momentum tracking in `bfloat16`, which can be expensive for training giant models. One potential solution is to factorize the momentum to save memory.

Table 11: One-shot evaluation on English NLP tasks. TriviaQA, NQs, and WebQs are NLG tasks and the rest are NLU tasks. This corresponds to Table 3 in the main text.

| Task | 1.1B | | 2.1B | | 7.5B | | 6.7B | 8B |
| | Adafactor | Lion | Adafactor | Lion | Adafactor | Lion | GPT-3 | PaLM |
| --- | --- | --- | --- | --- | --- | --- | --- | --- |
| #Tokens | | | 300B | | | | 300B | 780B |
| TriviaQA (EM) | 21.5 | **25.1** | 32.0 | **33.4** | 47.9 | **48.8** | 44.4 | 48.5 |
| NQs (EM) | 4.3 | **4.8** | 6.3 | **7.3** | **12.3** | 12.1 | 9.8 | 10.6 |
| WebQs (EM) | **7.5** | 6.3 | 8.4 | **8.7** | 12.1 | **13.3** | 15.1 | 12.6 |
| HellaSwag | **50.7** | 50.3 | **59.4** | 59.3 | 68.2 | **68.3** | 66.5 | 68.2 |
| StoryCloze | **74.8** | 74.4 | 78.2 | **78.3** | 81.2 | **81.5** | 78.7 | 78.7 |
| Winograd | 75.1 | **80.2** | 81.3 | **82.1** | **85.3** | 84.2 | 84.6 | 85.3 |
| Winogrande | 59.7 | **60.5** | 64.8 | **65.7** | **71.4** | 71.0 | 65.8 | 68.3 |
| RACE-m | **52.0** | 50.8 | **55.1** | 53.8 | 59.1 | **61.3** | 54.7 | 57.7 |
| RACE-h | **36.8** | 35.4 | 40.3 | **40.7** | **44.5** | 43.9 | 44.3 | 41.6 |
| PIQA | 69.4 | **69.9** | 71.3 | **72.1** | **75.5** | 74.5 | 76.3 | 76.1 |
| ARC-e | **64.3** | 62.0 | **69.5** | 68.9 | 72.4 | **72.7** | 62.6 | 71.3 |
| ARC-c | 31.2 | **32.9** | 37.3 | **38.0** | **43.3** | 42.6 | 41.5 | 42.3 |
| OpenbookQA | 44.8 | **48.0** | 48.4 | **49.0** | 51.4 | **52.4** | 53.0 | 47.4 |
| BoolQ | 54.3 | **56.7** | **64.1** | 62.9 | 73.5 | **73.9** | 68.7 | 64.7 |
| Copa | 75.0 | **78.0** | 83.0 | **84.0** | 85.0 | **87.0** | 82.0 | 82.0 |
| RTE | **55.6** | 52.4 | 49.8 | **59.2** | **63.9** | 62.5 | 54.9 | 57.8 |
| WiC | **47.6** | 47.3 | 46.1 | **48.1** | **50.9** | 48.1 | 50.3 | 47.3 |
| Multirc (F1a) | 35.9 | **44.3** | 45.0 | **48.8** | 44.7 | **59.2** | 64.5 | 50.6 |
| WSC | **76.5** | 75.4 | **79.6** | 79.3 | **86.7** | 85.6 | 60.6 | 81.4 |
| ReCoRD | 73.4 | **73.7** | **77.8** | 77.7 | 81.0 | **81.1** | 88.0 | 87.8 |
| CB | **46.4** | 44.6 | **48.2** | 44.6 | **51.8** | 46.4 | 33.9 | 41.1 |
| ANLI R1 | **33.3** | 30.1 | **32.4** | 31.2 | 31.5 | **34.0** | 31.6 | 32.4 |
| ANLI R2 | 29.8 | **31.8** | 29.8 | **30.6** | **32.4** | 31.9 | 33.9 | 31.4 |
| ANLI R3 | 29.8 | **31.8** | 31.4 | **31.9** | 33.6 | **34.2** | 33.1 | 34.5 |
| Avg NLG | 11.1 | **12.1** | 15.6 | **16.5** | 24.1 | **24.7** | 23.1 | 23.9 |
| Avg NLU | 53.2 | **53.9** | 56.8 | **57.4** | 61.3 | **61.7** | 58.5 | 59.4 |

Table 12: Hyperparameters for all the experiments.

| Model | Dropout | Stoch Depth | Augmentations | Optimizer | $\beta_1$ | $\beta_2$ | $lr$ | $\lambda$ |
|---|---|---|---|---|---|---|---|---|
| Train from scratch on ImageNet | | | | | | | | |
| ResNet-50 | - | - | - | AdamW | 0.9 | 0.999 | $3e-3$ | 0.1 |
| | | | | Lion | 0.9 | 0.99 | $3e-4$ | 1.0 |
| Mixer-S/16 | - | 0.1 | - | AdamW | 0.9 | 0.999 | $1e-2$ | 0.3 |
| | | | | Lion | 0.9 | 0.99 | $3e-3$ | 1.0 |
| Mixer-B/16 | - | 0.1 | - | AdamW | 0.9 | 0.999 | $1e-2$ | 0.3 |
| | | | | Lion | 0.9 | 0.99 | $3e-3$ | 3.0 |
| ViT-S/16 | 0.1 | 0.1 | - | AdamW | 0.9 | 0.999 | $1e-2$ | 0.1 |
| | | | | Lion | 0.9 | 0.99 | $1e-3$ | 1.0 |
| | - | - | RandAug: 2, 15 Mixup: 0.5 | AdamW | 0.9 | 0.999 | $3e-3$ | 0.1 |
| | | | | Lion | 0.9 | 0.99 | $3e-4$ | 1.0 |
| ViT-B/16 | 0.1 | 0.1 | - | AdamW | 0.9 | 0.999 | $3e-3$ | 0.3 |
| | | | | Lion | 0.9 | 0.99 | $1e-3$ | 1.0 |
| | - | - | RandAug: 2, 15 Mixup: 0.5 | AdamW | 0.9 | 0.999 | $1e-3$ | 1.0 |
| | | | | Lion | 0.9 | 0.99 | $1e-4$ | 10.0 |
| CoAtNet-1 | - | 0.3 | RandAug: 2, 15 Mixup: 0.8 | AdamW | 0.9 | 0.999 | $1e-3$ | 0.05 |
| | | | | Lion | 0.9 | 0.99 | $2e-4$ | 1.0 |
| CoAtNet-3 | - | 0.7 | RandAug: 2, 15 Mixup: 0.8 | AdamW | 0.9 | 0.999 | $1e-3$ | 0.05 |
| | | | | Lion | 0.9 | 0.99 | $2e-4$ | 1.0 |
| Pre-train on ImageNet-21K | | | | | | | | |
| ViT-B/16 | 0.1 | 0.1 | - | AdamW | 0.9 | 0.999 | $1e-3$ | 0.1 |
| | | | | Lion | 0.9 | 0.99 | $1e-4$ | 0.3 |
| ViT-L/16 | 0.1 | 0.1 | - | AdamW | 0.9 | 0.999 | $1e-3$ | 0.3 |
| | | | | Lion | 0.9 | 0.99 | $1e-4$ | 1.0 |
| Pre-train on JFT | | | | | | | | |
| ViT-B/16 | - | - | - | AdamW | 0.9 | 0.999 | $6e-4$ | 0.1 |
| | | | | Lion | 0.9 | 0.99 | $1e-4$ | 0.3 |
| ViT-L/16 | - | - | - | AdamW | 0.9 | 0.999 | $3e-4$ | 0.1 |
| | | | | Lion | 0.9 | 0.99 | $1e-4$ | 0.3 |
| ViT-H/14 | - | - | - | AdamW | 0.9 | 0.999 | $3e-4$ | 0.1 |
| | | | | Lion | 0.9 | 0.99 | $3e-5$ | 0.3 |
| ViT-g/14 & ViT-G/14 | - | - | - | Adafactor | 0.9 | 0.999 | $8e-4$ | 0.03 |
| | | | | Lion | 0.9 | 0.99 | $3e-5$ | 0.3 |
| Vision-language contrastive learning | | | | | | | | |
| LiT-B/∗-B | - | - | - | AdamW | 0.9 | 0.999 | $1e-3$ | - |
| | | | | Lion | 0.9 | 0.99 | $3e-4$ | |
| LiT-g/14-L | - | - | - | AdamW | 0.9 | 0.999 | $1e-3$ | 0.1 |
| | | | | Lion | 0.9 | 0.99 | $2e-4$ | 0.5 |
| BASIC-L | - | - | - | Adafactor | 0.9 | 0.999 | $5e-4$ | 0.01 |
| | | | | Lion | 0.9 | 0.99 | $2e-4$ | 0.1 |
| Diffusion model | | | | | | | | |
| Imagen base & super-resolution | - | - | - | AdamW | 0.9 | 0.999 | $1e-3$ | - |
| | | | | Lion | 0.9 | 0.99 | $1e-4$ | |
| Image generation on ImageNet | $64 \times 64$: 0.1 $128 \times 128$ & $256 \times 256$: 0.2 | - | - | AdamW | 0.9 | 0.999 | $3e-4$ | 0.01 |
| | | | | Lion | 0.9 | 0.99 | $3e-5$ | 0.1 |
| Autoregressive & masked language modeling | | | | | | | | |
| Small & Medium (PG-19, C4) & Large | - | - | - | AdamW | 0.9 | 0.99 | $3e-3$ | - |
| | | | | Lion | 0.95 | 0.98 | $3e-4$ | |
| Medium (Wiki-40B) | - | - | - | AdamW | 0.9 | 0.99 | $3e-3$ | 0.001 |
| | | | | Lion | 0.95 | 0.98 | $3e-4$ | 0.01 |
| 1.1B & 2.1B | - | - | - | Adafactor | 0.9 | 0.99 | $2e-3$ | 0.0005 |
| | | | | Lion | 0.95 | 0.98 | $2e-4$ | 0.005 |
| 7.5B | - | - | - | Adafactor | 0.9 | 0.99 | $1e-3$ | 0.001 |
| | | | | Lion | 0.95 | 0.98 | $1e-4$ | 0.01 |
| Language model fine-tuning | | | | | | | | |
| T5-Base & Large & 11B | 0.1 | - | - | AdamW | 0.9 | 0.99 | $3e-5$ | - |
| | | | | Lion | 0.95 | 0.98 | $3e-6$ | |

