# OpenReview forum: "Symbolic Discovery of Optimization Algorithms"
_NeurIPS.cc/2023/Conference — NeurIPS 2023 poster_

### Official Review · Reviewer_Y4eN · 2023-07-05

**Soundness:** 4 excellent
**Presentation:** 3 good
**Contribution:** 3 good
**Rating:** 6
**Confidence:** 5

**Summary:**

The paper introduces Lion, an automatic search strategy for deep learning optimizers. The paper claims that the resulting optimizer is more compute efficient than the baselines and achieves better accuracy on a variety of tasks from vision and language and model architectures. The search uses an evolutionary algorithm initialized wih AdamW that gradually mutates a population of optimizers and contains several additional strategies to increase search efficiency eg the removal of redundant instructions and a hash-based caching as well as funnel selection to mitigate overfitting on the proxy task. The standout discovery of the Lion optimizer is the use of a sign function in combination with a specific momentum strategy. Throughout their experiments, the authors demonstrate significant improvements, especially on larger architectures with large datasets.

**Strengths:**

 - The evaluation is very thorough. The authors include several vision and language models, contrastive learning as well as diffusion models. Throughout the experiments on Transformer models, I found the results to be a significant improvement over AdamW, especially since there has not really been a significant upgrade to AdamW in recent years. I am not fully convinced by the results on ResNets and I also think that it would have been better to have a larger selection of convolutional neural networks. But given the importance of transformers in ML, I think this is still a good result.
 - In general, I find the idea of the paper to automatically search for an optimizer to be relevant.
 - The presentation is clear overall.

**Weaknesses:**

 - I think the contribution is mostly technical and seems mostly like an engineering effort as the actual search strategy seems relatively simple.  For me, the results are still interesting enough for acceptance.
 - There is no code available and a lot of experiments are done on Google internal datasets such as JFT and the Imagen dataset which makes it impossible to reproduce a lot of the results in the paper. I could also not find any search hyperparameters in the appendix, eg learning rate used during the search.
 - The search method still produces quite a lot of unnecessary statements that need to be manually removed to arrive at the final program. While I understand the complexity of the task, I would still prefer a method that produces cleaner code automatically. Being able to produce simplified optimizers during training might even improve results, for example, what would happen if one would start the search again starting at the Lion optimizer instead of AdamW.


**Questions:**

 - Program 4 is an automatic simplification of Program 8. However, the authors further do manual simplifications to arrive at the final optimizer in program 1. I understand that the sign simplification (from the code in red) might be hard to automatically find, however, the method also includes a step to remove statements with minimal effect. Why does this removal step not remove the arcsin and clip functions if the authors manually found that their removal does not cause a quality drop?
 - The authors discuss that Lion needs a smaller learning rate due to the sign operation compared to AdamW. How does this lr compare to the one used during the search? Did you use a small learning rate during the search which might bias the search towards the sign operation or does the search actually find an optimizer that is optimal for a learning rate that is orders of magnitude smaller than the one used during the search process? In the second case, do you think that searching with a smaller lr might influence the results?
 - Would it be possible to do a similar search for a second-order method, eg initialized with Newton's method or do you think the additional complexity of the hessian integration would make the search space too complex?

**Limitations:**

On some tasks, the improvements are rather minimal. Also, the set of instructions is rather limited and does not allow for complex structures like for-loops. All significant limitations are addressed by the authors in the Appendix.

---

> ### Author Rebuttal · Authors · 2023-08-10
>
> Thank you for your constructive and insightful comments!
>
> > Program 4 is an automatic simplification of Program 8. However, the authors further do manual simplifications to arrive at the final optimizer in program 1. I understand that the sign simplification (from the code in red) might be hard to automatically find, however, the method also includes a step to remove statements with minimal effect. Why does this removal step not remove the arcsin and clip functions if the authors manually found that their removal does not cause a quality drop?
>
> This is a great question. We found two reasons why such statements remain in the program. (1) Some of the statements are added by evolution because it was helpful in some period of the search, for example, the clip function could help when the optimizers in the population are unstable and introduces very large gradients but becomes less helpful when the optimizer becomes inherently stable; (2) some statements are added because it helps on proxy tasks but makes no difference on larger target tasks, which can be seen as a type of meta-overfitting as discussed in section 2.3 and Figure 11; (3) some statements makes no difference in quality but are still preserved due to noise in evolution; note that since it doesn't make the quality worse, there is no pressure for evolution to remove them.
>
>
> > The authors discuss that Lion needs a smaller learning rate due to the sign operation compared to AdamW. How does this lr compare to the one used during the search? Did you use a small learning rate during the search which might bias the search towards the sign operation or does the search actually find an optimizer that is optimal for a learning rate that is orders of magnitude smaller than the one used during the search process? In the second case, do you think that searching with a smaller lr might influence the results?
>
> The learning rate is represented as a searchable constant in the program and also mutated during the evolutionary search. The initial value for the learning rate is selected based on the initial program (AdamW) so there should be no bias to push it towards the sign operation. In fact, we are quite surprised by the use of the sign operation in the discovered program since prior works using sign operation usually achieve worse results than Adam. Note that, as a baseline, we also performed a hyperparameter search for AdamW on the proxy tasks with more compute and the final result is still worse than the discovered optimizer.
>
> > Would it be possible to do a similar search for a second-order method, e.g. initialized with Newton's method or do you think the additional complexity of the hessian integration would make the search space too complex?
>
> Thanks for the suggestions! Since we use programs as the representation, it is definitely possible and quite natural to expand the search space to include more operations that can compose second order methods. Since this is our first attempt in this direction, we decided to keep it simple and only include operations for first order optimizers. One potential challenge with searching second order methods would be that introducing hessian matrices or operations like outer products can make the programs more expensive to evaluate, thus increasing the compute cost.

---

### Official Review · Reviewer_weMp · 2023-07-06

**Soundness:** 3 good
**Presentation:** 3 good
**Contribution:** 3 good
**Rating:** 6
**Confidence:** 3

**Summary:**

This paper proposes to formulate algorithm discovery as program search and discover optimization algorithms for deep neural network training. To bridge the large generalization gap between proxy and target tasks, the authors introduce program selection and simplification strategies. The proposed method discovers a simple and effective optimization algorithm called Lion, which is more memory-efficient than Adam as it only keeps track of the momentum. Lion is compared with widely used optimizers, such as Adam and Adafactor, for training a variety of models on different tasks. Empirical results demonstrate that Lion significantly outperforms these optimizers. The authors also examine the limitations of Lion and identify scenarios where its improvements are small or not statistically significant.

**Strengths:**

- The proposed method is clear and easy to understand
- Empirical results are very sufficient to demonstrate the superiority of proposed method


**Weaknesses:**

- The computation resources required by the proposed search method is unaffordable for most research groups
- Theoretical analysis is missing in this work


**Questions:**

Despite its sound empirical performances, I would also like to see some theoretical analysis on the searched Lion optimizer. How it can converge under which conditions? The authors may consider referring to [7] and see if similar convergence results can be obtained.

**Limitations:**

I suppose the key limitation of this work is its huge demand on computation power. For most research groups, the value of this work is solely the Lion optimizer (which is rather simple) instead of the search method, though it is certainly good to see some new optimizers no matter how they are found (by human or by machine).

---

> ### Author Rebuttal · Authors · 2023-08-10
>
> Thank you for your constructive and insightful comments!
>
> > The computation resources required by the proposed search method is unaffordable for most research groups
>
> We acknowledge that the search cost of Lion (3K TPU V2 days) is still significant, and improving its efficiency is an important future work. However, the search process is a one-time task, while the benefits of Lion are persistent and sustainable. In our evaluation of image classification, we pre-train ViT-G/14 and CoAtNet-7 on JFT, each requires 512 TPU V4 chips for one week, equivalent to 3K TPU V4 days, or approximately 12K TPU V2 days. Lion's ability to save over 2x compute on this task compared to Adafactor already offsets the search cost. As more researchers use Lion on various tasks, these compute savings will continue to grow.
>
> Furthermore, Lion can also bring benefits to production models, such as ads, which require frequent training and operate at a large scale. In these cases, the advantages of Lion can be even greater and outweigh the one-time search cost by a significant margin.
>
> > Theoretical analysis is missing in this work. Despite its sound empirical performances, I would also like to see some theoretical analysis on the searched Lion optimizer. How it can converge under which conditions? The authors may consider referring to [7] and see if similar convergence results can be obtained.
>
> We would like to emphasize that Lion is discovered through symbolic program search without any prior favoring sign operation or the decoupled momentum. While we acknowledge the importance of theoretical analysis, it is out of the scope of the current paper that focuses on the automatic discovery of the algorithm. Therefore, we leave it for future research endeavors.

---

### Official Review · Reviewer_Gizf · 2023-07-06

**Soundness:** 3 good
**Presentation:** 3 good
**Contribution:** 3 good
**Rating:** 6
**Confidence:** 4

**Summary:**

In this paper, a new optimizer is designed by program search. The search process overcomes the problem of distribution differences for small data set proxies and large tasks. The new optimizer converges faster and at the same time has strong generalization over a wide range of models for different tasks.

**Strengths:**

1. Apply the idea of program search of AutoML-Zero to the field of optimizer design. Efficient and highly generalized optimizers are designed.
2. Solved the problem of complex and sparse search space by abstract execution pruning, warm-start and proxy task.
3. solved the problem of distribution difference between proxy tasks and larger tasks by funnel selection.
4. Sufficient experiments. The experiments include the validation of extensive models on different tasks.

**Weaknesses:**

1. The original algorithm obtained from the search is still more complex. It needs expert knowledge and labor cost for simplification. See the simplification of Program 4 to Program 1 for details.
2. No specific validation experiments or theoretical analysis of the high memory efficiency of Lion compared to adam. It is only outlined at L233-L239. The authors can use some metrics, such as memory peaks, to compare the memory footprint of different optimizers.
3. The design for the search space is heuristic. In some places, the authors just make statements about the approach, which makes it difficult to understand. For example, the last of the three forms of mutation. Why is the hyperparameter in the statement modified instead of replacing the statement itself?

**Questions:**

See weaknesses.

**Limitations:**

Limitations have been well discussed

---

> ### Author Rebuttal · Authors · 2023-08-10
>
> Thank you for your constructive and insightful comments!
>
> > The original algorithm obtained from the search is still more complex. It needs expert knowledge and labor cost for simplification. See the simplification of Program 4 to Program 1 for details.
>
> We would like to point out that a large part of the simplification procedure is already automated. For example, program 4 (12 statements) is a result of automatic simplification from program 8 (21 statements). This automatic simplification is done through pruning with abstract execution (More details in Section 2.2 and Appendix I). Note that we only perform the manual simplification step on the best algorithms from the funnel selection and such simplification follows naturally from our analysis of the programs to understand it better.
>
> We agree with the reviewer that the simplification could be further automated by introducing heuristics for the mathematical equivalent transformation and ablation study, and leave it for future work.
>
> > No specific validation experiments or theoretical analysis of the high memory efficiency of Lion compared to adam. It is only outlined at L233-L239. The authors can use some metrics, such as memory peaks, to compare the memory footprint of different optimizers.
>
> Lion keeps track of only the momentum while Adam keeps track of both momentum and second moments. So we believe the memory efficiency is evident since there are less floats to keep in the memory. We agree with the reviewer that metrics memory peaks can be helpful but it depends on specific implementations and hardwares, so we leave the comprehensive comparison across different codebases and hardwares for future work.
>
> > The design for the search space is heuristic. In some places, the authors just make statements about the approach, which makes it difficult to understand. For example, the last of the three forms of mutation. Why is the hyperparameter in the statement modified instead of replacing the statement itself?
>
> The third type of mutation (modifying a statement by randomly changing its arguments) is introduced to make it easier to add small changes. Compared to replacing the existing statement with one randomly generated statement, i.e., a replacement mutation, it preserves most of the structure and allows "fine-tuning" the arguments. For example, consider a statement where most parts of the statements are good but one argument, which is a float constant, needs to be tuned. In this case, the modification mutation makes it easier to mutate the value of the constant based on the current value while a full replacement would make it harder to make such small changes. Additionally, the replacement mutation can already be composed with the first two types of mutations (insertion and deletion) so we didn't introduce it as a separate one. We will update the draft to make the motivation clearer.

---

### Official Review · Reviewer_hEv6 · 2023-07-26

**Soundness:** 3 good
**Presentation:** 3 good
**Contribution:** 3 good
**Rating:** 6
**Confidence:** 5

**Summary:**


Summary: The authors' work undoubtedly stands as an exceptional and noteworthy contribution to the field, showcasing a plethora of remarkable achievements that have the potential to significantly influence and shape the landscape of optimization and extend their impact well beyond its boundaries. As I delve into the intricacies of their research, it becomes evident that their dedication and innovative approach have yielded results that are both impressive and promising.

However, amidst the numerous commendable aspects of their study, there also exist certain primary concerns that warrant attention and resolution. While I acknowledge the impressive strides made by the authors, it is essential to critically address certain aspects that could further enhance the credibility and applicability of their findings. In the following detailed comments, I will highlight specific areas that require closer scrutiny, offering constructive feedback and suggestions to fortify the integrity and comprehensiveness of their research.



**Strengths:**

I am elated to express my joy and satisfaction upon the efficiency of the authors' method has transcended that of the renowned Adam optimizer, exhibiting its superiority across various tasks. This notable achievement undoubtedly highlights the ingenuity and prowess of the authors in devising such a remarkable approach.

The profound impact of their work is vividly demonstrated through the astonishing results presented in Table 1 and Figure 1, which undoubtedly leave one in awe. In particular, the Discovered optimizer, aptly named "Lion," showcased in Program 1, stands as a testament to the authors' ability to craft a solution that is both simplistic and elegant, evoking admiration from peers and fellow researchers alike.

Moreover, the decision made by the authors to simplify Program 4 and refine it into the more efficient Program 1 is not only logical but also reinforces the practicality of their approach. This thoughtful choice not only streamlines the methodology but also adds to the elegance and cohesiveness of their research.

In their diligent pursuit of showcasing the sparsity of the high-performance optimizer, the authors undertook a remarkably extensive and rigorous experimental endeavor, involving the execution of a staggering 2 million randomly selected programs. With meticulous precision and analytical acumen, they thoroughly assessed the performance of each program, leaving no stone unturned in their quest for a superior alternative to AdamW. Upon meticulously scrutinizing the results obtained from this monumental experiment, the authors arrived at a rather intriguing and significant revelation: none of the 2 million randomly selected programs exhibited a performance that surpassed the prowess of AdamW. This particular finding stands as a testament to the exceptional capabilities and efficiency of AdamW as a high-performance optimizer, solidifying its position as a prominent and leading choice in the optimization landscape. The sheer scale and comprehensiveness of this experimental undertaking are truly commendable, underscoring the authors' unwavering commitment to pursuing scientific rigor and thoroughness. Their methodical approach and the impressive volume of data analyzed serve as an invaluable resource for the scientific community, contributing to a deeper understanding of optimizer performance and fostering future advancements in the realm of optimization algorithms.

The authors' insightful use of meta-validation to select programs with superior generalization ability is indeed a reasonable and effective approach. Especially noteworthy is their ingenious technique of progressively increasing the complexity of the meta-validation task, aptly termed "Funnel selection." This strategic refinement undoubtedly enhances the efficiency of the selection process, allowing for the identification of programs with enhanced generalization capabilities.

Additionally, the thoughtful incorporation of sign updates and regularization is highly justified, especially when considering the context of adversarial attacks, where techniques like L_inf PGD attacks often rely on such sign update mechanisms. In fact, sign updates have also been previously employed as optimizers in prior works.

Undoubtedly, the analysis presented in Section 3.2 is both captivating and intellectually stimulating, offering valuable insights that resonate with the broader research community. The robustness and trustworthiness of these findings are indeed commendable and a testament to the authors' thoroughness in their investigations.

One fascinating observation that delights me is the correlation between model size and Lion's validation accuracy. The authors' remark on the positive relationship between increased model size and enhanced validation accuracy raises intriguing possibilities. The plausible explanation that the sign update technique increases uncertainty and ultimately improves the model's generalization ability opens exciting avenues for future research.


**Weaknesses:**


While their research presents impressive insights and advancements, it is essential to address one concern that arises from the potential handcrafted nature of the simplification and derivation process for the programs. A more elegant solution would involve automating this process as an inherent aspect of defining the search space, further enhancing the overall methodology and augmenting its potential impact.


Sign updates have also been previously employed as optimizers in prior works.


It is worth kindly noting that, when the proposed method is combined with more potent data augmentations, the relative performance gains might appear less pronounced. This insightful observation highlights the need for further exploration and optimization to fully unlock the method's potential under varying experimental conditions. While considering this aspect, I must admit that I personally feel a bit disappointed.

I am genuinely interested in exploring the application of the Lion algorithm to the domain of Stable Diffusion. It is pertinent to acknowledge that training stable diffusion models can be an extremely resource-intensive endeavor, often incurring substantial costs (for instance, approximately 1 million dollars for stable diffusion v1). Given this context, the potential of the proposed Lion method to be utilized in conjunction with stable diffusion holds tremendous appeal and significance. If the Lion algorithm proves to be compatible with stable diffusion models, it would indeed be a highly advantageous and desirable outcome. The ability to leverage the proposed method's capabilities in optimizing stable diffusion models could potentially mitigate the significant expenses associated with their training, thereby opening up new avenues for research and applications in this area. The prospect of such synergy between Lion and Stable Diffusion sparks a sense of optimism and excitement, as it has the potential to drive transformative advancements in the field. As such, I eagerly look forward to any insights or findings that may shed light on the feasibility and practicality of applying the Lion algorithm to the challenging task of stable diffusion, as this would undoubtedly represent a valuable and impactful contribution to the scientific community.


One of the  concerns in the field of AutoML revolves around the efficacy of the search process, causing a great deal of apprehension among researchers and practitioners. More precisely, a central aspect of this concern is rooted in the correlation between the performance exhibited by programs during the search phase, specifically on the small proxy, and their subsequent performance on the final task. This particular correlation is viewed with heightened attention, as it holds significant implications for the overall effectiveness of the AutoML approach. For instance, in the context of Neural Architecture Search (NAS), where the exploration of different architectural configurations plays a crucial role in achieving optimal results, there is a palpable sense of unease regarding architecture ranking. How well a particular architecture performs on the small proxy tasks might not always be indicative of its actual performance when applied to more extensive and state-of-the-art tasks, and thus, researchers seek to establish a robust link between the two domains. In attempts to address this pressing concern, the authors of the present paper have diligently employed a comprehensive set of measures. Their goal is to carefully select algorithms that can demonstrate a remarkable ability to generalize from small proxy tasks to larger and more complex real-world tasks, thereby enhancing the reliability and applicability of the AutoML approach. However, despite the earnest efforts made by the authors, there remains a notable omission in their work. They have not provided a quantitative analysis that establishes the explicit correlation between the performance achieved on the small proxy tasks and the corresponding performance on the final, real-world tasks. This apparent absence of a rigorous and quantitative examination is indeed regrettable, as such analytical insights are vital for instilling greater confidence in the effectiveness and reliability of the proposed AutoML techniques. In conclusion, while significant strides have been taken in addressing concerns regarding the search effectiveness in AutoML, the lack of a robust quantitative analysis linking the performance between the small proxy and the final tasks leaves a critical aspect unanswered. Efforts to bridge this analytical gap would undoubtedly enhance the overall understanding and trustworthiness of the AutoML methodologies employed.



**Questions:**


See *Weaknesses.

**Limitations:**


No. See *Weaknesses.

---

> ### Author Rebuttal · Authors · 2023-08-10
>
> Thank you for your constructive and insightful comments!
>
> > A more elegant solution would involve automating this process as an inherent aspect of defining the search space, further enhancing the overall methodology and augmenting its potential impact.
>
> We would like to point out that a large part of the simplification procedure is already automated. For example, program 4 (12 statements) is a result of automatic simplification from program 8 (21 statements). This automatic simplification is done through pruning with abstract execution (More details in Section 2.2 and Appendix I). Note that we only perform the manual simplification step on the best algorithms from the funnel selection and such simplification follows naturally from our analysis of the programs to understand it better.
>
> We agree with the reviewer that the simplification could be further automated by introducing heuristics for the mathematical equivalent transformation and ablation study, and we leave it for future work.
>
> > Sign updates have also been previously employed as optimizers in prior works.
>
> We have discussed the previous works that share similarities, such as the use of sign operation, with the discovered algorithms (Section 3.2 and Appendix K). Note that none of the prior works that uses sign update showed superior results compared to Adam and the ablation study in Appendix L demonstrated the benefit from the two linear interpolations used in Lion.
>
> > As such, I eagerly look forward to any insights or findings that may shed light on the feasibility and practicality of applying the Lion algorithm to the challenging task of stable diffusion, as this would undoubtedly represent a valuable and impactful contribution to the scientific community.
>
> In section 4.3, we evaluate Lion's performance in training diffusion models on ImageNet across three resolutions: 64, 128, and 256. While this approach differs from latent diffusion, it offers encouraging evidence of Lion's performance in stable diffusion.
>
> > More precisely, a central aspect of this concern is rooted in the correlation between the performance exhibited by programs during the search phase, specifically on the small proxy, and their subsequent performance on the final task. ... Efforts to bridge this analytical gap would undoubtedly enhance the overall understanding and trustworthiness of the AutoML methodologies employed.
>
> We agree with the reviewer that the generalization gap between the proxy tasks and the final / target tasks poses a great challenge. We discussed this challenge in section 2.3. Specifically, Figure 11 in the Appendix demonstrates the meta-overfitting issue caused by the gap. Programs that perform well on proxy tasks don't necessarily perform well on larger tasks. We alleviate this challenge through the funnel selection process described in section 2.3 that uses progressively larger tasks to further select out the programs that are more likely to work on the final task.

---

### Comment · Area_Chair_156V · 2023-08-21
**Message from AC**

Dear Authors,

Thanks for your responses to the reviews. I wanted to let you know that even tho not all the reviewers directly responded, your rebuttal will be accounted for during the final decision stage.

Best,
Your AC

---

### Decision · Program_Chairs · 2023-09-21

**Decision:**

Accept (poster)

**Comment:**

This paper proposes a symbolic search algorithm to discover novel optimizers. As a direct product of the algorithm, an novel optimizer is discovered, which is evaluated extensively on a wide range of large scale benchmarks, and show favorable performance than the standard AdamW optimizer. Reviewers are generally very positive about the findings, and one reviewer (hEv6) is particularly enthusiastic about it. The AC agrees that this is a solid work, and the fact that the optimizer demonstrating convincing performance gain over AdamW alone justifies acceptance. The AC agrees with Reviewer weMp that a deeper understanding of the discovered optimizer would be interesting, but also acknowledges that this may be beyond the scope of this work.